

# Development of a 2-way coupled ocean-wave model: assessment on a global $\mathrm{NEMO(v3.6)\text{-}WW3(v6.02)}$ coupled configuration

Xavier Couvelard[1], Florian Lemarié[2], Guillaume Samson[3], Jean-Luc Redelsperger[1], Fabrice Ardhuin[1], Rachid Benshila[4], and Gurvan Madec[5,2]

[1]Univ. Brest, CNRS, IRD, Ifremer, Laboratoire d'Océanographie Physique et Spatiale (LOPS), IUEM, Brest, France
[2]Univ. Grenoble Alpes, Inria, CNRS, Grenoble INP, LJK, 38000 Grenoble, France
[3]Mercator Océan, Toulouse, France
[4]LEGOS, University of Toulouse, CNES, CNRS, IRD, UPS, Toulouse, France
[5]Sorbonne Universités (UPMC, Univ Paris 06)-CNRS-IRD-MNHN, LOCEAN Laboratory, Paris, France

**Correspondence:** Florian Lemarié (florian.lemarie@inria.fr)

**Abstract.** This paper describes the implementation of a coupling between a three-dimensional ocean general circulation model (NEMO) and a wave model (WW3) to represent the interactions of the upper oceanic flow dynamics with surface waves. The focus is on the impact of such coupling on upper-ocean properties (temperature and currents) and mixed-layer depths (MLD) at global eddying scales. A generic coupling interface has been developed and the NEMO governing equations and boundary

conditions have been adapted to include wave-induced terms following the approach of McWilliams et al. (2004) and Ardhuin et al. (2008). In particular, the contributions of Stokes-Coriolis, Vortex and surface pressure forces have been implemented on top of the necessary modifications of the tracer/continuity equation and turbulent closure scheme (a 1-equation TKE closure here). To assess the new developments, we perform a set of sensitivity experiments with a global oceanic configuration at $1/4^o$ resolution coupled with a wave model configured at $1/2^o$ resolution. Numerical simulations show a global increase of

wind-stress due to the interaction with waves (via the Charnock coefficient) particularly at high latitudes, resulting in increased surface currents. The modifications brought to the TKE closure scheme and the inclusion of a parameterization for Langmuir turbulence lead to a significant increase of the mixing thus helping to deepen the MLD. This deepening is mainly located in the Southern Hemisphere and results in reduced sea-surface currents and temperatures.

## 1 Introduction

An accurate representation of ocean surface waves has long been recognized as essential for a wide range of applications ranging from marine meteorology to ocean and coastal engineering. Waves also play an important role in the short-term forecasting of extratropical and tropical cyclones by regulating sea-surface roughness (Janssen, 2008; Chen and Curnic, 2015; Hwang, 2015). More recently, the impact of waves on the oceanic circulation at global scale has triggered interest from the research and operational community (e.g. Hasselmann, 1991; Rascle and Ardhuin, 2009; D'Asaro et al., 2014; Fan and Griffies,

2014; Li et al., 2016; Law Chune and Aouf, 2018). In particular, surface waves are important for an accurate representation of air-sea interactions and their effect on fluxes of mass, momentum and energy through the wavy boundary layer must be





taken into account in ocean-atmosphere coupled models. For example, the momentum flux through the air-sea interface has traditionally been parameterized using the near surface winds (typically at 10 meter) and the atmospheric surface layer stability (Fairall et al., 2003; Large and Yeager, 2009; Brodeau et al., 2016). The physics of the coupling depends on the kinematics and

dynamics of the wave field. This includes a wide range of processes from wind-wave growth, nonlinear wave-wave interactions, wave-current interactions to wave dissipation. Such complex processes can only be adequately represented by a wave model.

Besides affecting the air-sea fluxes, waves define the mixing in the oceanic surface boundary layer (OSBL) via breaking and Langmuir turbulence. For example, Belcher et al. (2012) showed that Langmuir turbulence should be important over wide areas of the global ocean and more particularly in the Southern ocean. In this region, they show that the inclusion of the effect

of surface waves on the upper-ocean mixing during summertime allows for a reduction of systematic biases in the OSBL depth. Indeed their large eddy simulations (LES) suggest that under certain circumstances wave forcing can lead to large changes in the mixing profile throughout the OSBL and in the entrainment flux at the base of the OSBL. They concluded that wave forcing is always important when compared to buoyancy forcing, even in winter. Moreover, Polonichko (1997) and Van Roekel et al. (2012) emphasized the fact that the Langmuir cells intensity strongly depends on the alignment between the Stokes drift and

wind direction. Langmuir turbulence is maximum when wind and waves are aligned and becomes weaker as the misalignment becomes larger. Li et al. (2017) highlighted that ignoring the alignment of wind and waves (i.e. assuming that wind and waves are systematically aligned) in the Langmuir cells parameterizations leads to excessive mixing particularly in winter.

Most previous studies of the impact of ocean-wave interactions at global scale have been using an offline one-way coupling and included only parts of the wave-induced terms in the oceanic model governing equations (e.g. Breivik et al., 2015;

Law Chune and Aouf, 2018). In this study, the objective is to introduce a new online two-way coupled ocean wave modeling system with a great flexibility to be relevant for a large range of applications from climate modeling to regional short-term process studies. This modeling system is based on the Nucleus for European Modelling of the Ocean (NEMO, Madec, 2012) as the oceanic compartment and WAVEWATCH III® (hereinafter WW3, The WAVEWATCH III® Development Group, 2016) as the surface wave component. NEMO and WW3 are coupled using the OASIS Model Coupling Toolkit (OASIS3-MCT,

Van Roekel et al., 2012; Craig et al., 2017) which is widely used in the climate and operational communities. The various steps for our implementation are the following (*i*) inclusion of all wave-induced terms in NEMO, only neglecting the terms relevant for the surf zone which is outside the scope here (*ii*) modification of the NEMO subgrid scales physics (including the bulk formulation) to include wave effects and a parameterization for Langmuir turbulence (*iii*) development of the OASIS interface within NEMO and WW3 for the exchange of data between both models (*iv*) test of the implementation based on a realistic

global configuration at $1/4^o$ for the ocean and $1/2^o$ for the waves.

To go into the details of those different steps, the paper is organized as follows. The modifications brought to the oceanic model primitive equations, their boundary conditions, and the subgrid scales physics to account for wave-ocean interactions are described in Sec. 2. This includes the addition of the Stokes-Coriolis force, the Vortex force, the wave-induced pressure gradient. In Sec. 3 our modeling system coupling the NEMO oceanic model and the WW3 wave model via the OASIS3-MCT

coupler is described in details. Numerical simulations are presented in Sec. 4 using a global configuration at $1/4^o$ for the oceanic model and $1/2^o$ for the wave model. Using sensitivity runs, we assess those global configurations with particular emphasis on





the impact of wave-ocean interactions on mixed-layer depth, sea-surface temperature and currents, turbulent kinetic energy (TKE) injection, and kinetic energy. Finally, in Sec. 5, we summarize our findings and provide overall comments on the impact of two-way ocean-wave coupling in global configurations at eddy-permitting resolution.

## 2 Inclusion of wave-induced terms in the oceanic model NEMO

In order to set the necessary notations, we start by introducing the classical primitive equations solved by the NEMO ocean model. Note that between the two possible options to formulate the momentum equations, namely the so-called "vector invariant" and "flux" forms, we present here the first one which will be used for the numerical simulations in Sec. 4. With $\mathbf{u_h} = (u, v)$ the horizontal velocity vector, $\omega$ the dia-level velocity component, $\theta$ the potential temperature, $\rho$ the density, the Reynolds-averaged equations (with $\langle \cdot \rangle$ the averaging operator, omitted here for simplicity) are

$$\partial_t u = +(f+\zeta)v - \frac{1}{2}\partial_x \|\mathbf{u_h}\|^2 - \frac{\omega}{\mathrm{e}_3}\partial_k u - \frac{1}{\rho_0}\left(\partial_x(p_s + p_h) - \frac{(\partial_k p_h)(\partial_x z)}{\mathrm{e}_3}\right) - \frac{1}{\mathrm{e}_3}\partial_k \langle u'w' \rangle + F^u \tag{1}$$

$$\partial_t v = -(f+\zeta)u - \frac{1}{2}\partial_y \|\mathbf{u_h}\|^2 - \frac{\omega}{\mathrm{e}_3}\partial_k v - \frac{1}{\rho_0}\left(\partial_y(p_s + p_h) - \frac{(\partial_k p_h)(\partial_y z)}{\mathrm{e}_3}\right) - \frac{1}{\mathrm{e}_3}\partial_k \langle v'w' \rangle + F^v \tag{2}$$

$$\partial_t(\mathrm{e}_3\theta) = -\partial_x(\mathrm{e}_3\theta u) - \partial_y(\mathrm{e}_3\theta v) - \partial_k(\theta\omega) - \frac{1}{\mathrm{e}_3}\partial_k \langle \theta'w' \rangle + F^\theta \tag{3}$$

$$\partial_t \mathrm{e}_3 = -\partial_x(\mathrm{e}_3 u) - \partial_y(\mathrm{e}_3 v) - \partial_k\omega \tag{4}$$

$$\partial_k p = -\rho g \mathrm{e}_3 \tag{5}$$

Here $k$ is a non-dimensional vertical coordinate, lateral derivatives $\partial_x$ and $\partial_y$ have to be considered along the model coordinate, and $\mathrm{e}_3$ is the vertical scale factor given by $\mathrm{e}_3 = \partial_k z$, where $z$ is the local depth and $\rho$ is given by an equation of state (Roquet et al., 2015). The necessary boundary conditions include a kinematic surface and bottom boundary condition for the vertical velocity $w$

$$w(z=\eta) = \partial_t\eta + u|_{z=\eta}\,\partial_x\eta + v|_{z=\eta}\,\partial_y\eta, \qquad w(z=-H) = -u|_{z=-H}\,\partial_x H - v|_{z=-H}\,\partial_y H \tag{6}$$

with $\eta$ the height of the sea-surface, a momentum surface boundary condition for the Reynolds stress vertical terms

$$-\langle u'w' \rangle|_{z=\eta} = \frac{\tau_u^{\mathrm{oce}}}{\rho_0}, \qquad -\langle v'w' \rangle|_{z=\eta} = \frac{\tau_v^{\mathrm{oce}}}{\rho_0},$$

with $\boldsymbol{\tau}^{\mathrm{oce}} = (\tau_u^{\mathrm{oce}}, \tau_v^{\mathrm{oce}})$ the wind stress vector, and the dynamic boundary condition imposing the continuity of pressure at the air-sea interface. The kinematic boundary conditions (6) for $w(z=\eta)$ and $w(z=-H)$ translate into $\omega(z=\eta)=0$ and $\omega(z=-H)=0$. We do not include explicitly here the boundary conditions for the tracer equations since they are unchanged from classical primitive equations models in the presence of wave motions. As mentioned earlier, in equations (1) to (5) prognostic variables have to be interpreted in an Eulerian-mean sense even if the averaging operator is not explicitly included.

### 2.1 Modification of governing equations and boundary conditions

Asymptotic expansions of the wave effects based on Eulerian velocities (McWilliams et al., 2004) or Lagrangian mean equations (Ardhuin et al., 2008) lead to the same self-consistent set of equations for weak vertical current shears. These are





further applied and discussed by Uchiyama et al. (2010) and Bennis et al. (2011). The 3-component Stokes drift vector is $\mathbf{u}^s = (\widetilde{u}^s, \widetilde{v}^s, \widetilde{\omega}^s)$, and is non-divergent at lowest order (Ardhuin et al., 2008, 2017b). The coupled wave-current equations for the Eulerian mean velocity and tracers in a vector invariant form (the equivalent flux form is given in Appendix. A) are

$$
\begin{aligned}
\partial_t u = {} & +(f+\zeta)(v+\widetilde{v}^s) - \frac{1}{2}\partial_x\|\mathbf{u_h}\|^2 - \frac{(\omega+\widetilde{\omega}^s)}{\mathrm{e}_3}\partial_k u - \frac{\partial_x(p_s+\widetilde{p}^J)}{\rho_0} \\
& -\frac{1}{\rho_0}\left(\partial_x(p_h+\widetilde{p}^{\mathrm{FV}}) - \frac{(\partial_k(p_h+\widetilde{p}^{\mathrm{FV}}))(\partial_x z)}{\mathrm{e}_3}\right) + \frac{1}{\mathrm{e}_3}\partial_k\langle u'w'\rangle + F^u + \widetilde{F}^u \quad (7)
\end{aligned}
$$

$$
\begin{aligned}
\partial_t v = {} & -(f+\zeta)(u+\widetilde{u}^s) - \frac{1}{2}\partial_y\|\mathbf{u_h}\|^2 - \frac{(\omega+\widetilde{\omega}^s)}{\mathrm{e}_3}\partial_k v - \frac{\partial_y(p_s+\widetilde{p}^J)}{\rho_0} \\
& -\frac{1}{\rho_0}\left(\partial_y(p_h+\widetilde{p}^{\mathrm{FV}}) - \frac{(\partial_k p_h+\widetilde{p}^{\mathrm{FV}}))(\partial_y z)}{\mathrm{e}_3}\right) + \frac{1}{\mathrm{e}_3}\partial_k\langle v'w'\rangle + F^v + \widetilde{F}^v \quad (8)
\end{aligned}
$$

$$
\partial_t(\mathrm{e}_3\theta) = -\partial_x(\mathrm{e}_3\theta(u+\widetilde{u}^s) - \partial_y(\mathrm{e}_3\theta(v+\widetilde{v}^s)) - \partial_k(\theta(\omega+\widetilde{\omega}^s)) - \frac{1}{\mathrm{e}_3}\partial_k\langle\theta'w'\rangle + F^\theta \quad (9)
$$

$$
\partial_t\mathrm{e}_3 = -\partial_x(\mathrm{e}_3(u+\widetilde{u}^s)) - \partial_y(\mathrm{e}_3(v+\widetilde{v}^s)) - \partial_k(\omega+\widetilde{\omega}^s) \quad (10)
$$

$$
\partial_k p_h = -\rho g\mathrm{e}_3 - \partial_k\widetilde{p}^{\mathrm{FV}} + \rho_0\left(\widetilde{u}^s\partial_k u + \widetilde{v}^s\partial_k v\right) \quad (11)
$$

where wave-induced terms are represented with tildes. The extra contributions to the momentum equations include the Stokes-Coriolis force $\mathcal{W}_{\mathrm{St-Cor}}$, the vortex force $\mathcal{W}_{\mathrm{VF}}$, and a wave-induced pressure $\mathcal{W}_{\mathrm{Prs}}$

$$
\mathcal{W}_{\mathrm{St-Cor}} = \begin{pmatrix} f\widetilde{v}^s \\ -f\widetilde{u}^s \\ 0 \end{pmatrix}, \qquad
\mathcal{W}_{\mathrm{VF}} = \begin{pmatrix} \zeta\widetilde{v}^s - \frac{\widetilde{\omega}^s}{e_3}\partial_k u \\ -\zeta\widetilde{u}^s - \frac{\widetilde{\omega}^s}{e_3}\partial_k v \\ \frac{\widetilde{u}^s}{e_3}\partial_k u + \frac{\widetilde{v}^s}{e_3}\partial_k v \end{pmatrix}, \qquad
\mathcal{W}_{\mathrm{Prs}} = \begin{pmatrix} \widetilde{p}^J + \widetilde{p}^{\mathrm{FV}} \\ \widetilde{p}^J + \widetilde{p}^{\mathrm{FV}} \\ \widetilde{p}^{\mathrm{FV}} \end{pmatrix} \quad (12)
$$

where the terms involving horizontal derivatives of $\omega$ have been neglected in $\mathcal{W}_{\mathrm{VF}}$. In $\mathcal{W}_{\mathrm{Prs}}$, the $\widetilde{p}^J$ term corresponds to a depth uniform wave-induced kinematic pressure term, while $\widetilde{p}^{\mathrm{FV}}$ is a shear-induced three-dimensional pressure term associated with the vertical component of the vortex force. The vortex force contribution $\mathcal{W}_{\mathrm{VF}}$ can be further simplified by neglecting the terms involving the vertical shear as in Bennis et al. (2011), thus leading to $\mathcal{W}_{\mathrm{VF}}\cdot(0,0,1)^t = 0$ and $\widetilde{p}^{\mathrm{FV}} = 0$. This assumption has the advantage to leave the hydrostatic relation (11) unchanged. Our implementation of wave-induced terms in NEMO is inline with Bennis et al. (2011) and corresponds to the simplified form of (12)

$$
\mathcal{W}_{\mathrm{St-Cor}} = \begin{pmatrix} f\widetilde{v}^s \\ -f\widetilde{u}^s \\ 0 \end{pmatrix}, \qquad
\mathcal{W}_{\mathrm{VF}} = \begin{pmatrix} \zeta\widetilde{v}^s - \frac{\widetilde{\omega}^s}{e_3}\partial_k u \\ -\zeta\widetilde{u}^s - \frac{\widetilde{\omega}^s}{e_3}\partial_k v \\ 0 \end{pmatrix}, \qquad
\mathcal{W}_{\mathrm{Prs}} = \begin{pmatrix} \widetilde{p}^J \\ \widetilde{p}^J \\ 0 \end{pmatrix}.
$$

Regarding the joint modification of the tracers and continuity equations, it is clear that constancy preservation is maintained (i.e. a constant tracer field should remain constant during the advective transport) and that an additional wave related forcing must be added to the barotropic mode. The NEMO barotropic mode has been modified accordingly since the surface kinematic boundary condition (6) in terms of vertical velocities $w$ and associated $\widetilde{w}^s$ now reads

$$
w + \widetilde{w}^s = \partial_t\eta + (u|_{z=\eta} + \widetilde{u}^s|_{z=\eta})\partial_x\eta + (v|_{z=\eta} + \widetilde{v}^s|_{z=\eta})\partial_y\eta
$$





to express the fact that there is a source of mass at the surface that compensates the convergence of the Stokes drift, hence the barotropic mode is

$$
\begin{cases}
\partial_t \eta &= -\partial_x \left( (H+\eta)(\overline{u} + \overline{\widetilde{u}}^s) \right) - \partial_y \left( (H+\eta)(\overline{v} + \overline{\widetilde{v}}^s) \right) + P - E, \\
\partial_t \overline{u} &= +f\overline{v} - g\partial_x \eta - \dfrac{C_{b,x}}{(H+\eta)}\overline{u} + \overline{G^x} + \overline{\widetilde{G}^x} \\
\partial_t \overline{v} &= -f\overline{u} - g\partial_y \eta - \dfrac{C_{b,y}}{(H+\eta)}\overline{v} + \overline{G^y} + \overline{\widetilde{G}^y}
\end{cases}
\tag{13}
$$

where $\overline{\phi} = \frac{1}{H+\eta} \int_{-H}^{\eta} \phi dz$, $\mathbf{C}_b = (C_{b,x}, C_{b,y})$ the bottom drag coefficients, $\mathbf{G} = (\overline{G^x}, \overline{G^x})$ is the usual NEMO forcing term

containing coupling terms from the baroclinic mode as well as slowly varying barotropic terms (including nonlinear advective terms) held constant during the barotropic integration to gain efficiency. In (13), the $\overline{\widetilde{G}^x}$ and $\overline{\widetilde{G}^y}$ contain the additional wave-induced barotropic forcing terms corresponding to the vertical integral of the $\mathcal{W}_{\text{St-Cor}}$, $\mathcal{W}_{\text{Prs}}$, and $\mathcal{W}_{\text{VF}}$ terms which are also held constant during the barotropic integration. A thorough analysis on the impact of the additional wave-induced terms on energy transfers within an oceanic model can be found in Suzuki and Fox-Kemper (2016).

## 2.2   Computation and discretization of Stokes drift velocity profile

Reconstructing the full Stokes drift profiles in the ocean circulation model would require obtaining the surface spectra of the Stokes drift from the wave model. Instead, profiles are generally reconstructed considering a few important parameters, including the Stokes drift surface value $\mathbf{u}_{\text{h}}^{\text{s}}(\eta)$ and the norm of the Stokes volume transport $\|\mathbf{T}^s\|$. In Breivik et al. (2014) and Breivik et al. (2016), Stokes drift velocity profiles are derived under the deep-water approximation in the general form

$\mathbf{u}_{\text{h}}^{\text{s}}(z) = \mathbf{u}_{\text{h}}^{\text{s}}(\eta)\mathcal{S}(z, k_e)$ with $k_e$ a depth-independent spatial wavenumber chosen such that the norm of the depth integrated Stokes transport (assuming an ocean of infinite depth) is equal to $\|\mathbf{T}^s\|$. The functions $\mathcal{S}_{\text{B14}}(z, k_e)$ from Breivik et al. (2014) and $\mathcal{S}_{\text{B16}}(z, k_e)$ from Breivik et al. (2016) for $z \in [-H, \eta]$ are given by

$$
\mathcal{S}_{\text{B14}}(z, k_e) = \left( \frac{e^{2k_e(z-\eta)}}{1 - 8k_e(z-\eta)} \right), \qquad \mathcal{S}_{\text{B16}}(z, k_e) = e^{2k_e(z-\eta)} - \sqrt{2k_e \pi (\eta - z)}\,\text{erfc}(\sqrt{2k_e(\eta - z)}).
$$

with erfc the complementary error function. It can be easily shown that for an ocean of infinite depth, the vertical integral of

those functions are respectively equal to $\frac{1}{6k_e}$ for $\mathcal{S}_{\text{B16}}$ and $\frac{1.34089}{8k_e} \approx \frac{1}{5.97k_e}$ for $\mathcal{S}_{\text{B14}}$. Standard computations of Stokes drift in numerical models are done in a finite difference sense, however due to the fast decay of $\mathbf{u}_{\text{h}}^{\text{s}}(z)$ with depth, a finite volume approach seems more adequate, in this case

$$
(\mathbf{u}_{\text{h}}^{\text{s}})_k = \frac{\mathbf{u}_{\text{h}}^{\text{s}}(\eta)}{(\text{e}_3)_k} \int_{z_{k-1/2}}^{z_{k+1/2}} \mathcal{S}(z, k_e)\text{dz} = \frac{\mathbf{u}_{\text{h}}^{\text{s}}(\eta)}{(\text{e}_3)_k} \left[ \mathcal{I}(z_{k+1/2}, k_e) - \mathcal{I}(z_{k-1/2}, k_e) \right]
$$

The $\mathcal{S}_{\text{B16}}$ function is more adapted for this kind of approach since the primitive function does only require special functions

available in the fortran standard

$$
\mathcal{I}_{\text{B16}}(z, k_e) = \frac{1}{6k_e} \left[ e^{2k_e(z-\eta)} + 4k_e(z-\eta)\mathcal{S}_{\text{B16}}(z, k_e) \right]
$$



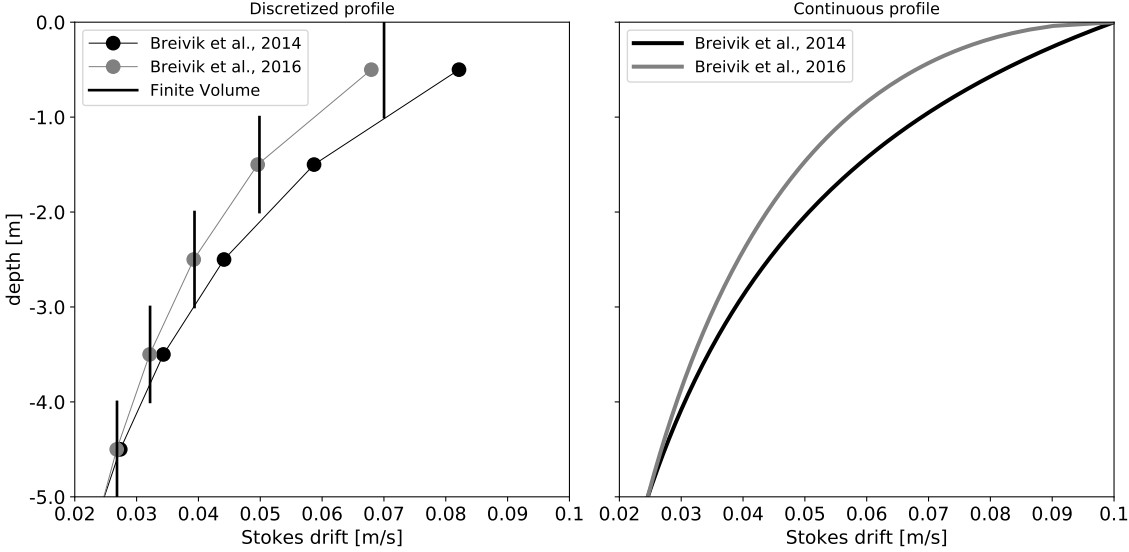

**Figure 1.** Left panel: reconstructed zonal component of Stokes drift profile for $\|\mathbf{T}^s\| = 0.4\,\mathrm{m^2\,s^{-1}}$, $u^s(z = \eta) = 0.1\mathrm{m\,s^{-1}}$, and $v^s(z = \eta) = 0\,\mathrm{m\,s^{-1}}$ for a 1 m resolution vertical grid using the Breivik et al. (2014) function (black dots), Breivik et al. (2016) function (grey dots), and the finite volume Breivik et al. (2016) function (black vertical lines). Right panel: their continuous counterparts.

Since NEMO is discretized on an Arakawa C-grid, the components of the Stokes drift velocity must be evaluated at cell interfaces, a simple average weighted by layer thicknesses is used :

$$\widetilde{u}^s_{i+1/2,j,k} = \frac{(\mathrm{e}_3)_{i,j,k} u^s_{i,j,k} + (\mathrm{e}_3)_{i+1,j,k} u^s_{i+1,j,k}}{2\,(\mathrm{e}_3)_{i+1/2,j,k}}, \qquad \widetilde{v}^s_{i,j+1/2,k} = \frac{(\mathrm{e}_3)_{i,j,k} v^s_{i,j,k} + (\mathrm{e}_3)_{i,j+1,k} v^s_{i+1,j,k}}{2\,(\mathrm{e}_3)_{i,j+1/2,k}}$$

Note that no explicit computation of the vertical component of the Stokes drift is necessary since in (7)-(11) $\widetilde{\omega}^s$ only appears summed to $\omega$ such that the relevant variable is $\omega + \widetilde{\omega}^s$ as a whole.

As illustrated in Fig. 1, for the typical vertical resolution used in most global models the properties of the discretized Stokes profiles can be very different from their continuous counterparts. Indeed, the $\mathcal{S}_{\mathrm{B16}}(z, k_e)$ has been considered superior to the $\mathcal{S}_{\mathrm{B14}}(z, k_e)$ because the vertical shear near the surface is expected to be better reproduced. However in Fig. 1 it is shown that

this is no longer the case at a discrete level since the discrete vertical gradients at one meter depth turns out to be larger for $\mathcal{S}_{\mathrm{B14}}(z, k_e)$ compared to $\mathcal{S}_{\mathrm{B16}}(z, k_e)$. In this case, the fast variations of $\mathcal{S}_{\mathrm{B16}}(z, k_e)$ near the surface can not be represented by the computational vertical grid.





### 2.3 Subgrid scales physics

#### 2.3.1 Turbulent kinetic energy prognostic equation and boundary conditions

Under the assumption of horizontal homogeneity, generally retained in general circulation models, the contribution from Stokes drift to the Turbulent Kinetic Energy (TKE) prognostic equation arises from the Vortex force vertical term $\mathcal{W}_{\mathrm{VF}}^z = \widetilde{u}^s \partial_z u + \widetilde{v}^s \partial_z v$ in the hydrostatic relation (11). Mimicking the way the TKE equation is usually derived (see e.g. Tennekes and Lumley, 1972) and using an averaging operator $\langle \cdot \rangle$ satisfying the "Reynolds properties", we find that the turbulent fluctuations, defined as $\phi' = \langle \phi \rangle - \phi$, $(\phi = p, \rho, u, v)$, associated with the $\mathcal{W}_{\mathrm{VF}}^z$ term are

$$(\mathcal{W}_{\mathrm{VF}}^z)' = \widetilde{u}^s \partial_z u' + \widetilde{v}^s \partial_z v'$$

after multiplication by $w'$ and averaging we obtain

$$\langle w'(\mathcal{W}_{\mathrm{VF}}^z)' \rangle = \widetilde{u}^s \partial_z \langle u'w' \rangle + \widetilde{v}^s \partial_z \langle v'w' \rangle - \widetilde{u}^s \langle u' \partial_z w' \rangle - \widetilde{v}^s \langle v' \partial_z w' \rangle$$

where the last two terms in the right-hand-side cancel with similar terms appearing when forming the equations for $\langle u' \partial_t u' \rangle$ and $\langle v' \partial_t v' \rangle$ (see eqn (A.7) and (A.8) in Skyllingstad and Denbo (1995)). The extra terms associated with the Stokes drift in 160 the horizontally homogeneous TKE equation are thus $u^s \partial_z \langle u'w' \rangle$ and $v^s \partial_z \langle v'w' \rangle$ which can be further rewritten as

$$\widetilde{u}^s \partial_z \langle u'w' \rangle = \langle u'w' \rangle \partial_z \widetilde{u}^s + \partial_z (\widetilde{u}^s \langle u'w' \rangle), \qquad \widetilde{v}^s \partial_z \langle v'w' \rangle = \langle v'w' \rangle \partial_z \widetilde{v}^s + \partial_z (\widetilde{v}^s \langle v'w' \rangle).$$

The first term will modify the shear production term, it can also be derived by taking the Lagrangian mean of the wave-resolved TKE equation (Ardhuin and Jenkins, 2006). The second will enter the TKE transport term which is usually parameterized as $-K_e \partial_z e$. The prognostic equation for the turbulent kinetic energy $e$ in NEMO under the assumption that $K_e = A^{vm}$, with 165 $A^{vm}$ the eddy viscosity, is thus

$$\partial_t e = \frac{A^{vm}}{\mathrm{e}_3^2} \left[ (\partial_k u)^2 + (\partial_k v)^2 + (\partial_k u)(\partial_k \widetilde{u}^s) + (\partial_k u^s)(\partial_k \widetilde{v}^s) \right] - A^{vt} N^2 + \frac{1}{\mathrm{e}_3} \partial_k \left[ \frac{A^{vm}}{\mathrm{e}_3} \partial_k e \right] - c_\epsilon \frac{e^{3/2}}{l_\varepsilon^2} \qquad (14)$$

with $A^{vt}$ the turbulent diffusivity, $N$ the local Brunt-Väisälä Frequency, $l_\varepsilon$ a dissipative length scale, and $c_\varepsilon$ a constant parameter (generally such that $c_\varepsilon \approx 1/\sqrt{2}$). Once the value of $e$ is know, eddy diffusivity/viscosity are given by

$$A^{vm} = C_m l_m \sqrt{e}, \qquad A^{vt} = A^{vm}/\mathrm{Prt}$$

with $\mathrm{Prt}$ the Prandtl number (see Sec. 10.1.3 in Madec (2012) for the detailed computation of $\mathrm{Prt}$), $l_m$ a mixing length scale, and $C_m$ a constant.

In addition to the modification of the shear production term in the TKE equation, the wave will affect the surface boundary condition both for $e$, $l_m$, and $l_\varepsilon$. The Dirichlet boundary condition traditionally used in NEMO for the TKE variable is modified into a Neumann boundary condition

$$\left( \frac{A^{vm}}{\mathrm{e}_3} \partial_k e \right)_{z=z_1} = -\rho_0 g \int\limits_0^{2\pi} \int\limits_0^\infty S_{\mathrm{ds}} d\omega d\theta = \Phi_{\mathrm{oce}} \qquad (15)$$





meaning that the injection of TKE at the surface is given by the dissipation of the wave field via the wave-ocean $S_{\mathrm{oce}}$ term, which is a sink term in the wave model energy balance equation usually dominated by wave breaking, converted into an ocean turbulence source term. In practice, this sum of $S_{\mathrm{ds}}$ is obtained as a residual of the source term integration, hence it also includes unresolved fluxes of energy to the high frequency tail of the wave model. Due to the placement at cell interfaces of

the TKE variable on the computational grid, the TKE flux is not applied at the free-surface but at the center of the top-most grid cell (i.e. at $z = z_1$). This amounts to interpret the half grid cell at the top as a constant flux layer which is consistent with the surface layer Monin-Obukhov theory.

The length scales $l_m$ and $l_\varepsilon$ are computed via two intermediate length scales $l_{\mathrm{up}}$ and $l_{\mathrm{dwn}}$ estimating respectively the maximum upward and downward displacement of a water parcel with a given initial kinetic energy. $l_{\mathrm{up}}$ and $l_{\mathrm{dwn}}$ are first initialized

to the length scale proposed by Deardorff (1980), $l_{\mathrm{up}}(z) = l_{\mathrm{dwn}}(z) = \sqrt{2e(z)/N^2(z)}$. The resulting length scales are then limited not only by the distance to the surface and to the bottom but also by the distance to a strongly stratified portion of the water column such as the thermocline. This limitation amounts to control the vertical gradients of $l_{\mathrm{up}}(z)$ and $l_{\mathrm{dwn}}(z)$ such that they are not larger that the variations of depth (Madec, 2012)

$$\partial_k |l.| \leq \mathrm{e}_3, \qquad l. = l_{\mathrm{up}}, l_{\mathrm{dwn}}$$

Then the dissipative and mixing length scale are given by $l_m = \sqrt{l_{\mathrm{up}} l_{\mathrm{dwn}}}$ and $l_\varepsilon = \min(l_{\mathrm{up}}, l_{\mathrm{dwn}})$. Following Redelsperger et al. (2001) (their Sec. 4.2.3), a boundary condition consistent with the Monin-Obukhov similarity theory for the length scale $l_{\mathrm{dwn}}$ (while $l_{\mathrm{up}}$ necessitates only a bottom boundary condition) is

$$l_{\mathrm{dwn}}(z = \eta) = \kappa \frac{(C_m c_\varepsilon)^{1/4}}{C_m} z_0$$

with $\kappa$ the von Karman constant and $C_m$, $c_\varepsilon$ the constant parameters in the TKE closure. The surface roughness length $z_0$ can

be directly estimated from the significant wave height provided by the wave model as $z_0 = 1.6 H_s$ (Rascle et al., 2008, their eqn (5)) which provides a proxy for the scale of the breaking waves. Note that in our study, no explicit parameterization of the mixing induced by near-inertial waves has been added (Rodgers et al., 2014). As highlighted by Breivik et al. (2015), without activating this *ad hoc* parameterization in the standard NEMO TKE scheme, the model does not mix deeply enough. They also speculated that this *ad hoc* mixing could mask effects of wave-related mixing processes such as Langmuir turbulence. For this

reason, it is thus not used in the present simulations.

### 2.3.2 Langmuir turbulence parameterization

Langmuir mixing is parameterized following the approach of Axell (2002). This parameterization takes the form of an additional source term $P_{\mathrm{LC}}$ in the TKE equation (14). $P_{\mathrm{LC}}$ is defined as

$$P_{\mathrm{LC}} = \frac{w_{\mathrm{LC}}^3}{d_{\mathrm{LC}}}$$





where $w_{LC}$ represents the vertical velocity profile associated with Langmuir cells and $d_{LC}$ their expected depth. Following Axell (2002), $w_{LC}$ and $d_{LC}$ are given by

$$w_{LC} = \begin{cases} c_{LC}\|\widehat{\mathbf{u}}^s_{LC}\|\sin\left(-\frac{\pi z}{d_{LC}}\right), & \text{if } -z \le d_{LC} \\ 0, & \text{otherwise} \end{cases} \quad , \quad -\int_{-d_{LC}}^{\eta} N^2(z)z\,dz = \frac{\|\widehat{\mathbf{u}}^s_{LC}\|^2}{2}$$

where $\|\widehat{\mathbf{u}}^s_{LC}\|$ is the portion of the surface Stokes drift contributing to Langmuir cells intensity and $c_{LC}$ a constant parameter. In the absence of information about the wave field it is generally assumed that $\|\widehat{\mathbf{u}}^s_{LC}\| \propto \|\boldsymbol{\tau}\|$. As mentioned in the introduction, Polonichko (1997) and Van Roekel et al. (2012) showed that the intensity of Langmuir cells is largely influenced by the angle between the Stokes drift and the wind direction. To reflect this dependency we account for this angle in our definition of $\|\widehat{\mathbf{u}}^s_{LC}\|$ via

$$\|\widehat{\mathbf{u}}^s_{LC}\| = \max\left\{\mathbf{u}^s(\eta)\cdot\mathbf{e}_{\boldsymbol{\tau}}, 0\right\}$$

with $\mathbf{e}_{\boldsymbol{\tau}}$ the unit vector in the wind-stress direction. Finally, a value for the parameter $c_{LC}$ must be chosen. Based on single-column experiments detailed in App. B, we find that parameter values in the range $0.15 - 0.3$ provide satisfactory results compared to LES simulations and will be considered for the numerical experiments discussed later in Sec. 4.2.

While the Axell (2002) parameterization was already implemented in NEMO there are three majors novelties in our implementation: *(i)* The online coupled strategy allows us to use the surface Stokes drift directly delivered by the wave model instead of the original value empirically estimated from the wind speed (e.g. 1.6% of the 10m wind) *(ii)* we only considered the component of the Stokes drift aligned with the wind and *(iii)* based on a series of single column simulations (see appendix B) the coefficients $c_{LC}$ evaluated to 0.15 by Axell (2002) is set up to a 0.3 value. Those changes together with the new surface boundary condition for the TKE equation, lead to a deeper penetration of the TKE inside the mixed layer and as shown in Sec. 4.2.3 greatly improved the MLD distribution.

## 3 Modeling system and coupling strategy

Our coupled model is based on the NEMO oceanic model, the WW3 wave model, and the OASIS library for data exchanges and synchronizations between both components.

### 3.1 Numerical models and coupling infrastructure

**The ocean model : NEMO**

NEMO is a state-of-the-art primitive-equation,split-explicit, free-surface oceanic model whose equations are formulated both in the vector invariant and flux forms (see (1) for the vector invariant form). The equations are discretized using a generalized vertical coordinate featuring, among others, the $z^*$-coordinate with partial step bathymetry and the $\sigma$-coordinate as well as a mixture of both (Madec, 2012). For efficiency and accuracy in the representation of external gravity waves propagation, model equations are split between a barotropic mode and a baroclinic mode to allow the possibility to adopt specific numerical





treatments in each mode. The NEMO equations are spatially discretized on an Arakawa C-grid in the horizontal and a Lorenz

grid in the vertical, and the time dimension is discretized using a Leapfrog scheme with a modified Robert-Asselin filter to damp the spurious numerical mode associated with Leapfrog (Leclair and Madec, 2009). For the current study the NEMO equations have been modified to include wave effects as described in (7) and (13). Moreover the modifications to the standard NEMO 1-equation TKE closure scheme are given in Sec. 2.3.

**The wave model : WW3**

The NEMO ocean model has been coupled to the WW3 wave model. In numerical models, waves are generally described using several phase and amplitude parameters. We provide here only the few sufficient details to understand the coupling of waves with the oceanic model, an exhaustive description of WW3 can be found in The WAVEWATCH III$^{®}$ Development Group (2016). WW3 integrates the wave action equation (Komen et al., 1994), with the spectral density of wave action $N_{\mathrm{w}}(k_{\mathrm{w}}, \theta_{\mathrm{w}})$, discretized in wavenumber $k_{\mathrm{w}}$ and wave propagation direction $\theta_{\mathrm{w}}$ for the spectral space (subscripts w are used here to avoid

confusion with previously introduced notations).

$$\partial_t N_{\mathrm{w}} + \partial_\phi \left( \dot{\phi} N_{\mathrm{w}} \right) + \partial_\lambda \left( \dot{\lambda} N_{\mathrm{w}} \right) + \partial_{k_{\mathrm{w}}} \left( \dot{k_{\mathrm{w}}} N_{\mathrm{w}} \right) + \partial_{\theta_{\mathrm{w}}} \left( \dot{\theta_{\mathrm{w}}} N_{\mathrm{w}} \right) = \frac{S}{\sigma}, \tag{16}$$

where $\lambda$ is longitude, $\phi$ is latitude, and $S$ is the net spectral source term that includes the sum of rate of change of the surface elevation variance due to interactions with the atmosphere via wind-wave generation and swell dissipation ($S_{\mathrm{atm}}$), nonlinear wave-wave interactions ($S_{\mathrm{nl}}$), and interactions with the upper ocean that is generally dominated by wave breaking ($S_{\mathrm{oc}}$). Those

parameterized source terms are important in waves-ocean coupling. Indeed, as shown earlier in (15) the $S_{\mathrm{oc}}$ term is used to compute the TKE flux transmitted to the ocean, and the $S_{\mathrm{in}}$ term enters in the computation of the wave-supported stress. They are here computed following Ardhuin et al. (2010b). In (16), the dot variables correspond to a propagation speed given by

$$\dot{\phi} = \left( c_g \cos\theta_{\mathrm{w}} + v|_{z=\eta} \right) R^{-1} \tag{17}$$

$$\dot{\lambda} = \left( c_g \sin\theta_{\mathrm{w}} + u|_{z=\eta} \right) (R\cos\phi)^{-1} \tag{18}$$

$$\dot{\theta_{\mathrm{w}}} = c_g \sin\theta_{\mathrm{w}} \tan\phi R^{-1} + \sin\theta_{\mathrm{w}} \frac{\partial\omega}{\partial\phi} - \frac{\cos\theta_{\mathrm{w}}}{\cos\phi} \frac{\partial\omega}{\partial\lambda} (k_{\mathrm{w}}R)^{-1} \tag{19}$$

$$\dot{k_{\mathrm{w}}} = -\frac{\partial\sigma}{\partial H} \frac{\mathbf{k}}{k_{\mathrm{w}}} \cdot \boldsymbol{\nabla}D - \mathbf{k} \cdot \boldsymbol{\nabla}\mathbf{u}_h(z=\eta), \tag{20}$$

where $R$ is earth radius, $\mathbf{u}_h(z=\eta) = (u|_{z=\eta}, v|_{z=\eta})$ are the surface currents provided by the ocean model, $c_g$ is the group velocity, $\omega$ the absolute radian frequency, and $H$ the mean water depth. Equation (16) is solved for each spectral component $(k_{\mathrm{w}}, \theta_{\mathrm{w}})$ which are coupled by the advection and source terms. Equations (17)-(20) show how the oceanic currents affect the

advection of the wave action density, there are also indirect effects via the source term (Ardhuin et al., 2009).

**The coupler : OASIS3-MCT**

The practical coupling between NEMO and WW3 has been implemented using the OASIS3-MCT (Valcke, 2012; Craig et al., 2017) software primarily developed for use in multi-component climate models. This software provides the tools to couple





various models at low implementation and performance overhead. In particular, thanks to MCT (Jacob et al., 2005), it includes
the parallelization of the coupling communications and runtime grid interpolations. For efficiency, interpolations are formulated
in the form of a matrix-vector multiplication where the matrix containing the mapping weights is computed offline one for all.
In practice, after compiling OASIS3-MCT, the resulting library is linked to the component models so that they have access to
the specific interpolation and data exchange subroutines. Now that we have described the different components involved in our
coupled system, we go into the details of the nature of the data exchanged between both models.

### 3.2  Oceanic surface momentum flux computation

Surface waves affect the momentum exchange between the ocean and the atmosphere in two different ways. First the modi-
fication of surface rugosity acts on the incoming atmospheric momentum flux $\boldsymbol{\tau}^{\mathrm{atm}}$. Second, even if most of the momentum
flux going into the waves is quickly transferred to the water column through wave breaking (we call this portion $\boldsymbol{\tau}^{\mathrm{oce}}$), a part
of it is consumed by the wave field and contributes to the growing waves (the so-called wave-supported stress). These two
coupling processes are taken into account in our coupled framework. The 10 meters wind $\mathbf{u}_{10}^{\mathrm{atm}}$ is sent to the wave model and
used to compute its own atmospheric wind-stress $\boldsymbol{\tau}_{\mathrm{ww3}}^{\mathrm{atm}}$ assuming neutral stratification, i.e. $\boldsymbol{\tau}_{\mathrm{ww3}}^{\mathrm{atm}} = \rho_a C_{\mathrm{DN}} \|\mathbf{u}_{10}^{\mathrm{atm}}\| \mathbf{u}_{10}^{\mathrm{atm}}$ with
$C_{\mathrm{DN}}$ a neutral drag coefficient. Then the wave model computes the momentum flux transferred to the ocean $\boldsymbol{\tau}_{\mathrm{ww3}}^{\mathrm{oce}}$ as well as
the dimensionless Charnock parameter $\alpha_{\mathrm{ch}}$ which characterizes the sea surface roughness (Charnock, 1955; Janssen, 2009).
Using the latest available values of $\alpha_{\mathrm{ch}}$, $\boldsymbol{\tau}_{\mathrm{ww3}}^{\mathrm{atm}}$, $\boldsymbol{\tau}_{\mathrm{ww3}}^{\mathrm{oce}}$, and $\mathbf{u}_{10}^{\mathrm{atm}}$, the oceanic model computes an atmospheric wind-stress $\boldsymbol{\tau}^{\mathrm{atm}}$
using its own bulk formulation and the local value of the momentum flux going into the water column is

$$\boldsymbol{\tau}^{\mathrm{oce}} = \boldsymbol{\tau}^{\mathrm{atm}} - \left(\boldsymbol{\tau}_{\mathrm{ww3}}^{\mathrm{atm}} - \boldsymbol{\tau}_{\mathrm{ww3}}^{\mathrm{oce}}\right) \tag{21}$$

where the $\boldsymbol{\tau}_{\mathrm{ww3}}$ quantities are interpolated from the wave grid to the oceanic grid. In NEMO, the wind-stress is computed using
the IFS[1] bulk formulation such as implemented in the AeroBulk[2] package (Brodeau et al., 2016). In particular the roughness
length which enters in the definition of the drag coefficient is function of the Charnock parameter $\alpha_{\mathrm{ch}}$

$$z_0 = \alpha_{\mathrm{ch}} \frac{u_\star^2}{g} + \alpha_m \frac{\nu}{u_\star}$$

where $\alpha_m = 0.11$, $u_\star$ is the friction velocity, and $\nu$ the air kinematic viscosity whose contribution is significant only asymp-
totically at very low wind speed. Note that in the uncoupled case the default value of the Charnock parameter is $\alpha_{\mathrm{ch}}^0 = 0.018$.
In our implementation, the momentum fluxes are computed using the absolute wind $\mathbf{u}_{10}^{\mathrm{atm}}$ at 10m rather than the relative wind
$\mathbf{u}_{10}^{\mathrm{atm}} - \mathbf{u}_h(z = \eta)$. Indeed, several recent studies have emphasized that the use of relative winds is relevant only when a full
coupling with an atmospheric model is available since in a forced mode it leads to an unrealistically large loss of oceanic eddy
kinetic energy (e.g. Renault et al., 2016).

Note that in our coupling strategy two different values of the atmospheric wind-stress and of the wave to ocean wind
stress are computed with two different bulk formulations. This strategy is not fully satisfactory since it breaks the momentum

---

[1]Integrated Forecasting System: https://www.ecmwf.int/en/forecasts/documentation-and-support/changes-ecmwf-model/ifs-documentation
[2]http://aerobulk.sourceforge.net/





| Variable | description | | units |
|---|---|---|---|
| $\mathbf{u}_h(z=\eta)$ | Oceanic surface currents | O→W | $\mathrm{m\,s^{-1}}$ |
| $\mathbf{u}_{10}^{\mathrm{atm}}$ | 10 m-winds from external dataset | O→W | $\mathrm{m\,s^{-1}}$ |
| $\mathbf{u}_h^s(z=\eta)$ | Sea-surface Stokes drift | W→O | $\mathrm{m\,s^{-1}}$ |
| $\|\mathbf{T}^s\|$ | norm of the Stokes drift volume transport | W→O | $\mathrm{m^2\,s^{-1}}$ |
| $\Phi_{\mathrm{oc}}$ | TKE surface flux multiplied by $\rho_0$ | W→O | $\mathrm{W\,m^{-2}}$ |
| $\alpha_{\mathrm{ch}}$ | Charnock parameter | W→O | - |
| $\boldsymbol{\tau}_{\mathrm{w}}^{\mathrm{ww3}}$ | Wave-supported stress | W→O | $\mathrm{N.m^{-2}}$ |
| $\widetilde{p}^J$ | wave-induced pressure | W→O | $\mathrm{m^2\,s^{-2}}$ |
| $H_s$ | Significant wave height | W→O | m |

**Table 1.** Variables exchanged between NEMO (O) and WW3 (W) via the OASIS3-MCT coupler. The 10 m wind $\mathbf{u}_{10}^{\mathrm{atm}}$ is interpolated online by WW3 and does not go through the OASIS3-MCT coupler.

conservation. However, it was necessary in practice since the WW3 results were very sensitive to the bulk formulation and at
the same time it was not conceivable to use the WW3 bulk formulation to force the ocean model because the latter ignores the effects of stratification in the atmospheric surface layer. Previous implementations in NEMO (e.g. Breivik et al., 2015; Alari et al., 2016; Staneva et al., 2017; Law Chune and Aouf, 2018) assumed that the wave field only acts on the norm of $\boldsymbol{\tau}^{\mathrm{atm}}$ and not on its orientation. Instead of (21), the atmospheric wind stress was corrected as follow:

$$\boldsymbol{\tau}^{\mathrm{oce}} = \boldsymbol{\tau}^{\mathrm{atm}} \left( \frac{\boldsymbol{\tau}_{\mathrm{ww3}}^{\mathrm{oce}}}{\boldsymbol{\tau}_{\mathrm{ww3}}^{\mathrm{atm}}} \right)$$

However, this approach potentially leads to artificially large values of $\boldsymbol{\tau}^{\mathrm{oce}}$ when $\boldsymbol{\tau}_{\mathrm{ww3}}^{\mathrm{atm}}$ is small and it does not take into account the slight change in $\boldsymbol{\tau}^{\mathrm{oce}}$ direction induced by the waves.

### 3.3 Additional details about the practical implementation

In Table 1 the different variables exchanged between the oceanic and wave models are given. All variables are 2D variables meaning that no 3D arrays are exchanged through the coupler. All 2D interpolation are made through a distance weighted
bilinear interpolations. The time discretization steps $\Delta t_{\mathrm{ww3}}$ for WW3 and $\Delta t_{\mathrm{nemo}}$ for NEMO are generally different with $\Delta t_{\mathrm{ww3}} > \Delta t_{\mathrm{nemo}}$ and chosen such that $\Delta t_{\mathrm{ww3}} = n_t \Delta t_{\mathrm{nemo}}$ ($n_t \in \mathbb{N}, n_t \geq 1$). In this case, coupling fields from NEMO to WW3 are averaged in time between two exchanges, while fields from WW3 to NEMO are sent every $\Delta t_{\mathrm{ww3}}$ steps and therefore updated every $n_t$ time steps in NEMO. If $\Delta t_{\mathrm{ww3}} > \Delta t_{\mathrm{nemo}}$, the coupler time-step is set to $\Delta t_{\mathrm{ww3}}$. Note that our current implementation does not include an explicit coupling between waves and sea-ice while it is known that waves lead to ice
break-up, pancake ice formation and associated enhancement of both freezing and melting and, in return, this wave dissipation in ice-covered water (e.g. Stopa et al., 2018) leads to ice drift. Such explicit coupling is currently under development within the NEMO framework (Boutin et al., 2019).





## 4 Global $1/4^o$ coupled wave-ocean simulations

### 4.1 Experimental setup and experiments

#### 4.1.1 The global coupled ORCA25 configuration

The wave hindcasts presented here are all based on the WW3 model in its version 6.02 configured with a single grid at $0.5^o$
resolution in longitude and latitude. A spectral grid with 24 directions and 31 frequencies exponentially spaced over the in-
terval $[f_{\min}, f_{\max}]$ with $f_{\min} = 0.037$ Hz and $f_{\max} = 0.7$ Hz. A one-step monotonic third-order coupled space-time advection
scheme (a.k.a. Ultimate Quickest scheme) is used with a specific procedure to alleviate the so-called garden sprinkler effect
(Tolman et al., 2002). As suggested in Phillips (1984), the dissipation induced by wave breaking is proportional to the local
saturation spectrum (see also Ardhuin et al., 2010a; Rascle and Ardhuin, 2013). The wind input growth rate at high frequencies
is based on the formulation of Janssen (1991) with an additional "sheltering" term to reduce the effective winds for the shorter
waves (Chen and Belcher, 2000; Banner and Morison, 2010). For the computation of nonlinear wave-wave interactions, the
discrete interaction approximation of Hasselmann et al. (1985) is used. This last approximation is known to be inaccurate but it
is thought that the associated error are usually compensated by a proper adjustment of the dissipation source term (Banner and
Young, 1994; Ardhuin et al., 2007). As mentioned earlier in Sec. 3.2, the model was run with 10 meter winds, without any air-
sea stability correction. No wave measurements were assimilated in the model but the stand-alone wave model was developed
based on spectral buoy and SAR data (Ardhuin et al., 2010b), and calibrated against altimeter data by adjusting the wind-wave
coupling parameter (Rascle and Ardhuin, 2013). The WW3 time-step for the global configurations is $\Delta t_{\text{ww3}} = 3600$ s.

For the oceanic component, we use a global ORCA025 configuration at a $1/4^o$ horizontal resolution (Barnier et al., 2006).
The vertical grid is designed with 75 vertical $z$-levels with vertical spacing increasing with depth. Grid thickness is about 1 m
near the surface and increases with depth to reach 200 m at the bottom. Partial steps are used to represent the bathymetry.
The LIM3 sea-ice model is used for the sea-ice dynamics and thermodynamics (Rousset et al., 2015). The vertical mixing
coefficients are obtained from the 1-equation TKE scheme described in Sec. 2.3 and the convective processes are mimicked
using an enhanced vertical diffusion parameterization which increases vertical diffusivity to $10 \text{ m}^2 \text{ s}^{-1}$ where static instability
occurs. Water density is computed from temperature and salinity through the use of a polynomial formulation of the UNESCO
(1983) non-linear equation of state (Roquet et al., 2015). The numerical options are the one commonly chosen by the Drakkar
group[3]. The vector-invariant form momentum advection is using Arakawa and Lamb (1981) for the vorticity and a specific
formulation to control the Hollingsworth instability (Ducousso et al., 2017). Momentum lateral viscosity is biharmonic and
acts along geopotential surfaces. It is set to a value of $1.5 \times 10^{11} \text{ m}^4.\text{s}^{-1}$. Advection of tracers is performed with a flux-
corrected-transport (FCT) scheme (Lévy et al., 2001), and lateral diffusion of tracers is harmonic and acts along iso-neutral
surface. It is set to a value of $300 \text{ m}^2 \text{ s}^{-1}$ at the equator. The bottom friction is non-linear and the lateral boundary condition is
free-slip. In this setup, the baroclinic time step is set to $\Delta t_{\text{nemo}} = 900$s, and a barotropic time step 30 times smaller. Compared

---

[3]https://www.drakkar-ocean.eu/





to the standard uncoupled ORCA025 configuration, the additional computational cost associated to WW3 and to the exchanges
through the coupler is about $20\%$.

### 4.1.2 Atmospheric forcings

The atmospheric fields used to force both ocean and wave models are based on the ECMWF (European Centre for Medium-
Range Weather Forecasts) ERA-Interim reanalysis (Dee et al., 2011). Corrections have been applied to better reproduce the
diurnal cycle of forcing fields and to guarantee that the ERA-Interim mean state for rainfalls, shortwave and longwave radiative
fluxes are consistent with satellite observations from Remote Sensing Systems (RSS) Passive Microwave Water Cycle (PMWC)
product (Hilburn, 2009) and GEWEX SRB 3.1 data[4]. Momentum and heat turbulent surface fluxes are computed using the IFS
bulk formulation from AeroBulk package (Brodeau et al., 2016) using air temperature and humidity at 2 meters, mean sea level
pressure and 10 meter winds.

### 4.1.3 Sensitivity experiments and objectives

Sensitivity experiments have been conducted to check the proper implementation of various components of present coupled
modelling system. For the sake of clarity, our developments are split in four components: *(i)* the modification of the wind-stress
by waves through the Charnock parameter and the inclusion of wave-supported stress, *(ii)* the modifications of the NEMO
governing equations through the Stokes-Coriolis, Vortex force and wave-induced surface pressure terms, *(iii)* the addition of
a Langmuir turbulence parameterization, and *(iv)* the modifications to the TKE scheme. As summarized in Tab. 2, sensitivity
experiments are designed in such a way to incrementally increase the level of complexity and test the effect of each component.

The $\mathrm{No\_CPL}$ experiment corresponds to the classical NEMO setup where wave effect is parameterized through a wind-stress
dependent TKE surface boundary condition as suggested by Craig and Banner (1994). In this approach, a Dirichlet surface
boundary condition is used and expressed as follow: $e(z=\eta) = \frac{1}{2}(15.8\alpha_{CB})^{2/3}\frac{\|\boldsymbol{\tau}^{\mathrm{atm}}\|}{\rho_0}$ with $\alpha_{CB} = 100$. The WS_CPL
expriment is identical as $\mathrm{No\_CPL}$ except that the wave coupling is introduced within the wind-stress computation, as described
in Sec. 3.2. ST_CPL experiment is as WS_CPL except that all terms relative to the Stokes drift described in Sec. 2.1 are added
in NEMO. All_CPL(1&2) experiments are like ST_CPL but with a fully modified TKE scheme including Langmuir cells
parameterization. All those simulations have been performed for 2 years (2013-2014) where 2013 is let as spinup and only
2014 is analysed. It must be clear that the objective here is not to go through a thorough physical analysis of coupled solutions
but to check and validate our numerical developments.

## 4.2 Numerical results

### 4.2.1 Waves impact on oceanic Wind stress

The wave distribution being inhomogeneous on the globe, it is expected that with the wave-modified wind stress parametriza-
tion the stress should follow more closely the wave patterns. In Fig. 2, the seasonal averages of the significant wave height

---

[4]http://gewex-srb.larc.nasa.gov/common/php/SRB_data_products.php



| Case | O-W coupling | Wave-supported stress + Charnock parameter | $\mathcal{W}_{St-Cor}$, $\mathcal{W}_{VF}$, $\mathcal{W}_{Prs}$ | Langmuir cells parameterization(rn_lc) | Modified TKE scheme |
|---|---|---|---|---|---|
| No_CPL | no | no | no | no | no |
| WS_CPL | 2-way | yes | no | no | no |
| ST_CPL | 2-way | yes | yes | no | no |
| All_CPL1 | 2-way | yes | yes | yes (0.15) | yes |
| All_CPL2 | 2-way | yes | yes | yes (0.30) | yes |

**Table 2.** Various model configurations analyzed in Sec. 4.2.

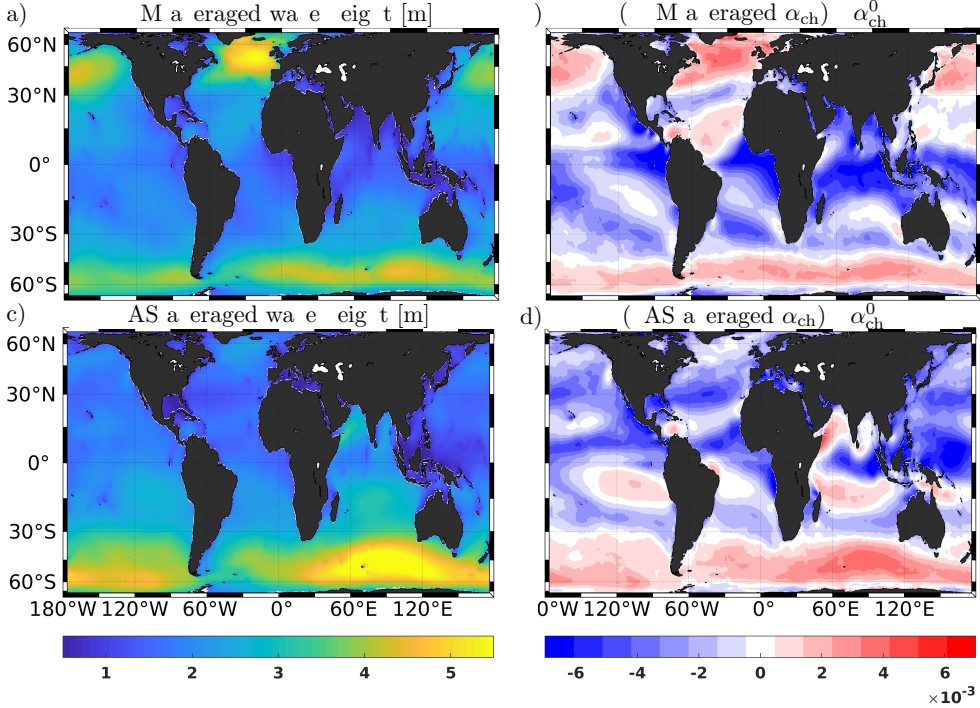

**Figure 2.** (a&c):Seasonal averaged Significant wave height (in meters) (b&d):Seasonal Averaged differences between the Charnock parameter as computed by the wave model and the default value $\alpha_{ch}^0 = 0.018$.

and of the differences between the Charnock coefficient computed by the wave model and the default constant value used in
the uncoupled case ($\alpha_{ch}^0 = 0.018$) are shown. As expected, the Charnock parameter tends to be stronger in the area where the
waves are the higher. Generally an increase of the Charnock parameter is observed in the northern and southern basin while
there is net decrease of $\alpha_{ch}$ near the equator. There is also a strong seasonality in the north hemisphere with a reduction in
summer and a strong increase in winter. The differences between $\alpha_{ch}$ and $\alpha_{ch}^0$ are very latitudinal with very few longitudinal
variations.





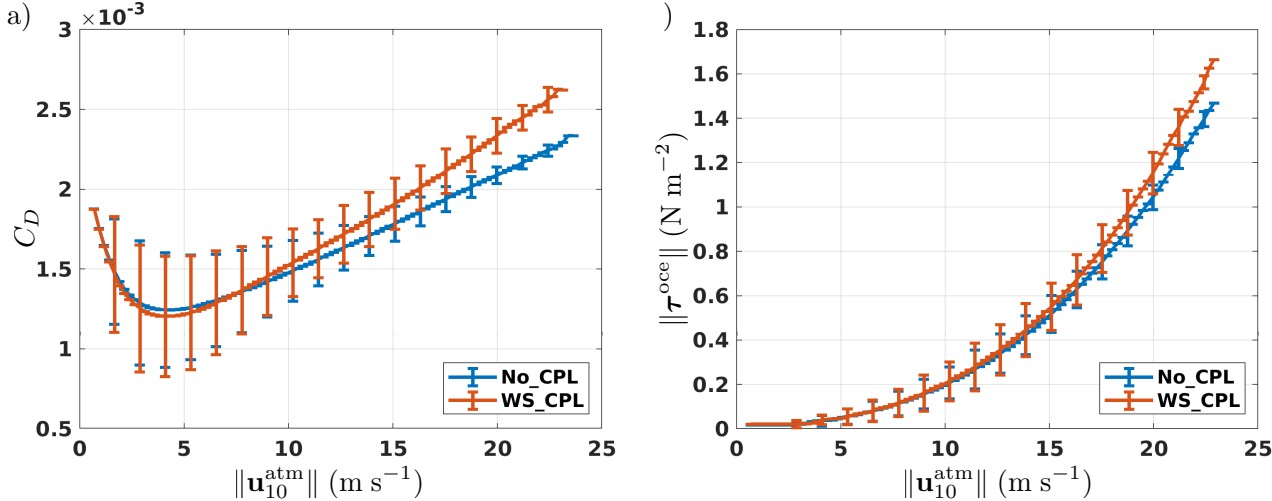

**Figure 3.** (a) Drag coefficient ($C_D$) as a function of the 10 meters wind speed $\|\mathbf{u}_{10}^{\mathrm{atm}}\|$ and (b) Wind Stress norm $\|\boldsymbol{\tau}^{\mathrm{oce}}\|$ as a function of $\|\mathbf{u}_{10}^{\mathrm{atm}}\|$ (Black curves represent the mean value while the vertical bars represent the standard deviation.)

To isolate the effect of the Charnock parameter we compare the results obtained in the No_CPL and WS_CPL experiments. Those two experiments show relatively similar sea surface temperature patterns meaning that the modifications of the wind-stress $\|\boldsymbol{\tau}^{\mathrm{oce}}\|$ between those two cases are primarily due to the use of different Charnock parameters and the inclusion of the wave-supported stress. Fig. 3 (panel a) illustrates that the Charnock parameter mostly affects the drag coefficient $C_D$, hence the surface wind-stress, for large winds. The ocean-wave coupling does not lead to appreciable differences in the drag coefficient

$C_D$ for wind speeds lower than $8\,\mathrm{m\,s}^{-1}$. On the contrary, since large values of the Charnock parameter are observed for large wind speeds, the coupling significantly increases the drag (as well as its variance) at high winds. Fig. 3 (panel b) shows how the wind-stress is modified by this increase of the drag coefficient jointly with the wave-supported stress which tends to decrease the wind-stress magnitude (Fig. 4). At low wind speed the wind-stress magnitude is not affected by the coupling with waves while for strong winds the increase of wind-stress associated with the increased drag coefficient is always larger than

the decrease associated to the wave-supported stress. This latter effect reduces the wind stress by no more than 2%, for the characteristic scales of our study, this correction is thus almost negligible. The wind-stress changes due to the coupling with waves seen in our simulations are very localized in time and space and it is thus difficult to conclude on their overall effect on the upper ocean dynamics such as the Ekman pumping and the surface currents.

### 4.2.2   Waves impact on surface TKE injection

As described in section 2.3, in the ocean-waves coupled case, the surface boundary condition for the TKE equation is a Neumann condition whose value is directly given by the wave model, unlike the uncoupled case where a Dirichlet condition is imposed. We aim here at assessing the impact on the order of magnitude of the near-surface TKE. Since the Neumann



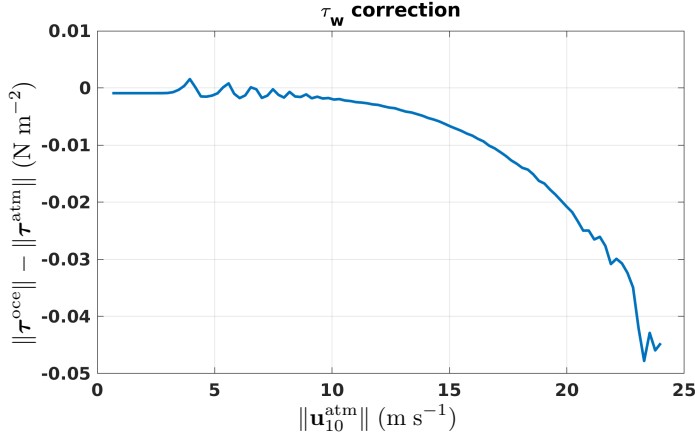

**Figure 4.** Wind-stress difference $\|\boldsymbol{\tau}^{\mathrm{oce}}\| - \|\boldsymbol{\tau}^{\mathrm{atm}}\|$ (N m$^{-2}$) due to the correction made for growing waves for the WS_CPL experiment, as a function of the 10 meters wind speed.

boundary condition is applied at the center of the top-most grid box (i.e. approximately at $50\,\mathrm{cm}$ depth), we compare in Fig. 5 the TKE value at 1 meter depth between the coupled (All_Cpl2) and the uncoupled (No_CPL) case. Positive values means that near-surface TKE is larger in the coupled simulation. It shows an almost homogeneous increase of the TKE (up to more than 100%) in the extra-tropical areas. While low seasonal variability in the extra-tropical areas is visible in Fig. 5, a spatial averaging by hemisphere (Fig. 6) highlights seasonal variability with a strong increase in both near-surface TKE value and TKE difference between both experiments during winter. In Fig. 5, 6 (and also in the reminder of the paper), the spatial averaging is made between $25\,\mathrm{S}$ and $60\,\mathrm{S}$ in the southern hemisphere and between $25\,\mathrm{N}$ and $60\,\mathrm{N}$ in the northern hemisphere to avoid any conflicts with sea-ice and to remove the equatorial region from the comparison. The increase of the surface TKE injection associated with waves is expected to contribute to an overall increase of mixed layer depth provided that the mixing length diagnosed by the turbulent closure scheme allows to effectively propagate this additional TKE deeper in the mixed layer.

### 4.2.3   Waves impact on Mixed layer depth

In this section, we evaluate the wave effect on vertical mixing using the mixed layer depth (MLD) as a relevant metric. Fig. 7 represents the seasonally averaged difference in MLD between the coupled (All_CPL2) and the uncoupled (No_CPL) case relative to the No_CPL case (i.e. $(h_{\mathrm{mld}}^{\mathrm{nocpl}} - h_{\mathrm{mld}}^{\mathrm{cpl}})/h_{\mathrm{mld}}^{\mathrm{nocpl}}$ with $h_{\mathrm{mld}}$ considered negative downward). It shows a significant deepening of the mixed layer at high latitudes in the coupled case with only very few localized mixed layer shallowing up to 60% mainly in the southern hemisphere. To assess whether the overall deepening of the mixed layer is realistic, we make a comparison with available observations. Fig. 8 represents the spatially averaged MLD where the blue line is the spatially averaged MLD obtained from ARGO floats (available during the same period) in both hemispheres. In the northern hemisphere (Fig. 8, a), there is only a slight improvement compared to data during winter and late summer when implementing the coupling with waves. In the southern hemisphere (Fig. 8, b) the situation is rather different. From January to July, the deepening of MLD

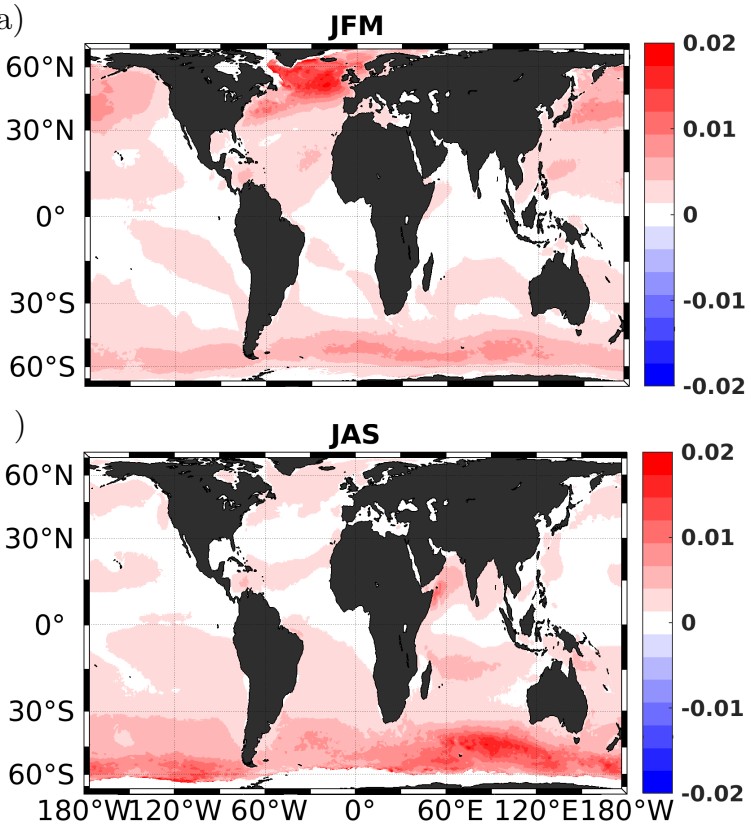

**Figure 5.** Seasonal differences of 1 meter depth turbulent kinetic energy ($\mathrm{m^2\,s^{-2}}$) between the coupled case (All_Cpl2) and the uncoupled case (No_CPL). (a) for January, February, March (JFM) and (b) for July, August, September (JAS).

induced by the wave coupling significantly reduces the bias between the model and ARGO data. From July to December, results in the coupled case show an overestimation of MLDs which were already too deep in the uncoupled case, therefore increasing

the bias between data and model. Since mesoscale activity make direct comparisons to data unreliable for such a short period of time, we compare the normalized distribution of MLD between the different simulations and available ARGO data. Results are presented in Fig. 9 for year 2014 (panel (a)) and during summer only (panel (b)). In both cases the improvements in the northern hemisphere are very modest. As far as the southern hemisphere is concerned the coupling with waves leads to a significant improvement compared to MLD derived from ARGO floats despite the fact that there are still too many low MLD values in

the range $50-100\,\mathrm{m}$. In comparison with the uncoupled case there is a more realistic spreading toward deeper mixed layer depths. More particularly in summer (Fig. 9, b), the probability density function (PDF) in the coupled case matches almost perfectly the ones computed from ARGO data. Despite the fact that we did not activate the *ad-hoc* extra mixing induced by near-inertial waves (Rodgers et al., 2014) our implementation of the wave-ocean interaction leads to a significant deepening of the MLD in a realistic way. To better understand which components of the wave-ocean coupling are responsible for this





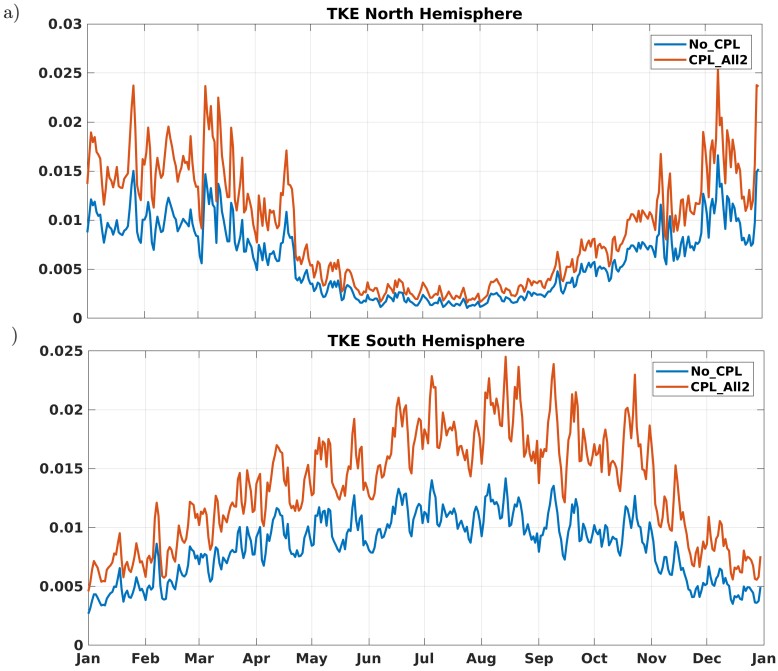

**Figure 6.** Spatially averaged turbulent kinetic energy ($\mathrm{m^2\,s^{-2}}$) at one meter depth over (a) the Southern Hemisphere and (b) the Northern Hemisphere.

improvement, the summer PDF in the South hemisphere has been computed for each of the experiments described in Tab. 2. Results are shown in Fig. 10. First of all, it can be seen that all the wave-ocean retroactions described in previous sections lead to an improvement in terms of mixed layer depth distribution compared to the uncoupled case. Indeed, the modification of the wind stress by the wave field introduced in WS_CPL, increases both surface currents and near surface TKE values resulting in a slight deepening of the MLD. Adding the Stokes drift related terms in the primitive equations contributes only modestly to the deepening of the MLD while most of the improvement results from the modified TKE scheme and more specifically from the Langmuir parameterization. It is somewhat reassuring to see that the better agreement with ARGO data is obtained when all components of the coupling are activated.

### 4.2.4 Waves impact on sea-surface temperature

Since the near-surface mixing is strengthened by the coupling we can expect an impact on sea-surface temperature (SST). Fig. 11 represents the time series of SST for each hemisphere. The Northern hemisphere is characterized by a warm bias during summer with a very slight improvement when coupling with waves. In the Southern hemisphere (Fig. 11, b) the summer warm bias is reduced by half in the coupled simulation and a slight warming occurs during the winter. While the summer surface cooling might be linked to the mixed-layer deepening, the winter warming might be rather linked to advection as observed by



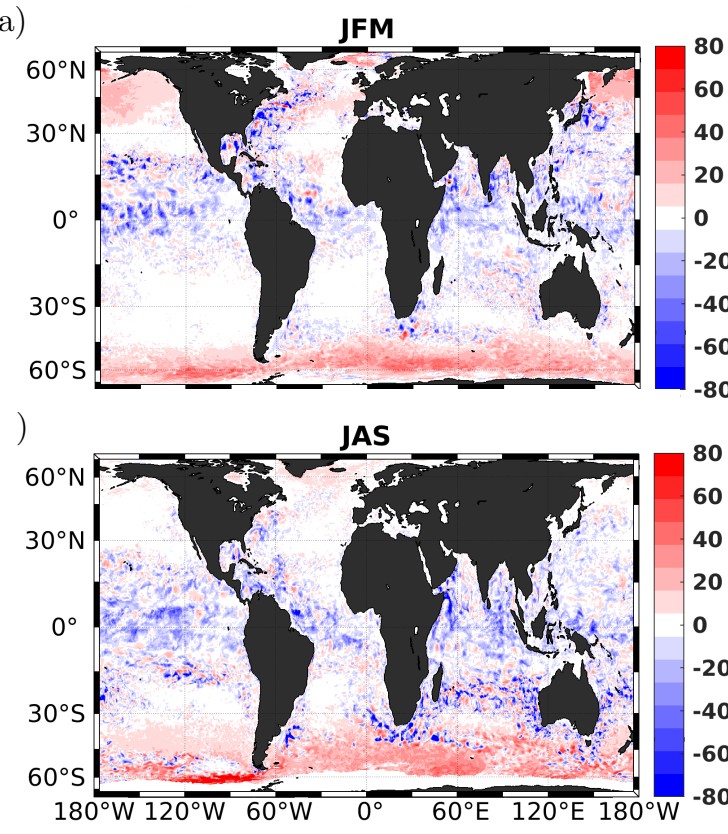

**Figure 7.** (a) & (b): Seasonaly averaged MLD differences (All_CPL2-No_CPL) relative to the uncoupled simulation No_CPL. Red color correspond to deeper MLD for All_CPL2.

Alari et al. (2016) for the Baltic sea. It could also result from an increased heat content during winter leading to higher SST

during summer. To better characterize the wave impact on the SST, we show in Fig. 12 (panel a) the difference in term of annual mean between the No_CPL experiment and OSTIA analysis exhibiting a cold bias in the No_CPL simulation in equatorial and tropical regions and a warm bias in the northern part of the Pacific ocean. The coupling with waves tends to diminish the cold bias (see Fig. 12, b) especially in the Pacific ocean and the warm bias in the north Pacific is significantly reduced. As already noticed by Law Chune and Aouf (2018) the warming in the equatorial and tropical regions mainly results from a lower

wind stress caused by a value of the Charnock parameter lower than the value used in the uncoupled case (see Fig. 2,b,d). A consequence is a decrease of the drag coefficient leading to smaller turbulent exchange coefficients reducing the heat flux. As mentionned above, in extra-tropical regions, some warm bias tend to be partially reduced by the extra mixing induced by the waves at high latitude or/and by the increased turbulent transfer coefficient. The tendency of the wave coupling to improve the near-surface temperature distribution can also be verified on a time-latitude Hovmuller diagram like the ones shown in Fig. 13.

For instance, it can be seen that the summer warm bias in the northern hemisphere (Fig. 13,a) coincides well with the cooling



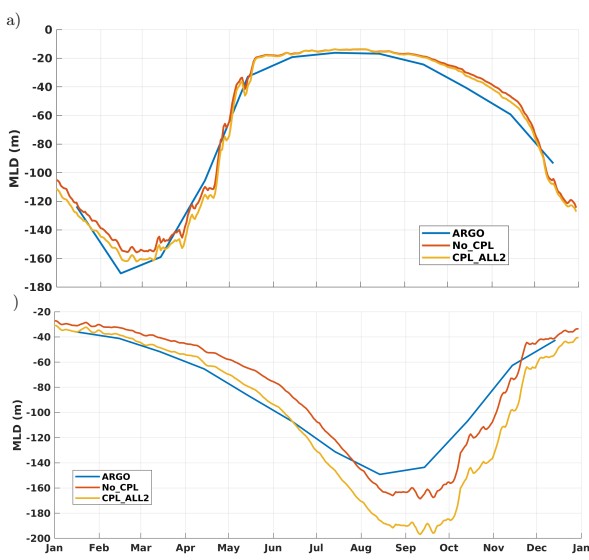

**Figure 8.** Spatially averaged MLD for (a) the north hemisphere and (b) the south hemisphere

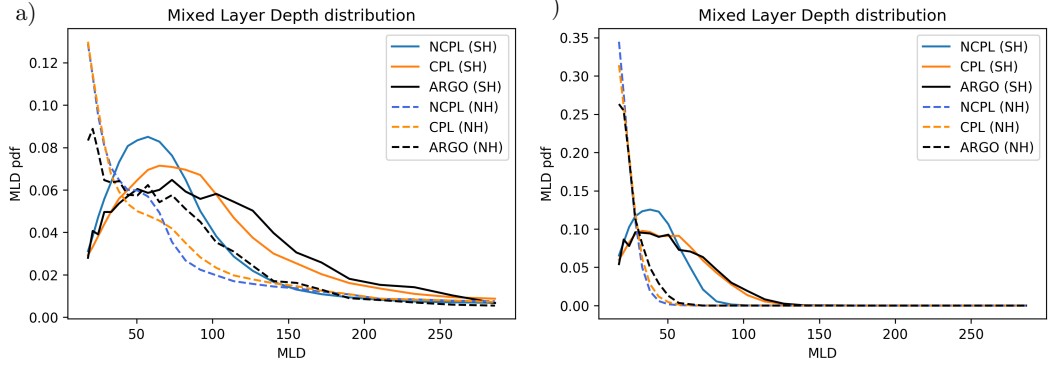

**Figure 9.** Mixed Layer Depth probability density function, for (a): the full 2014 year and (b): summer 2014.





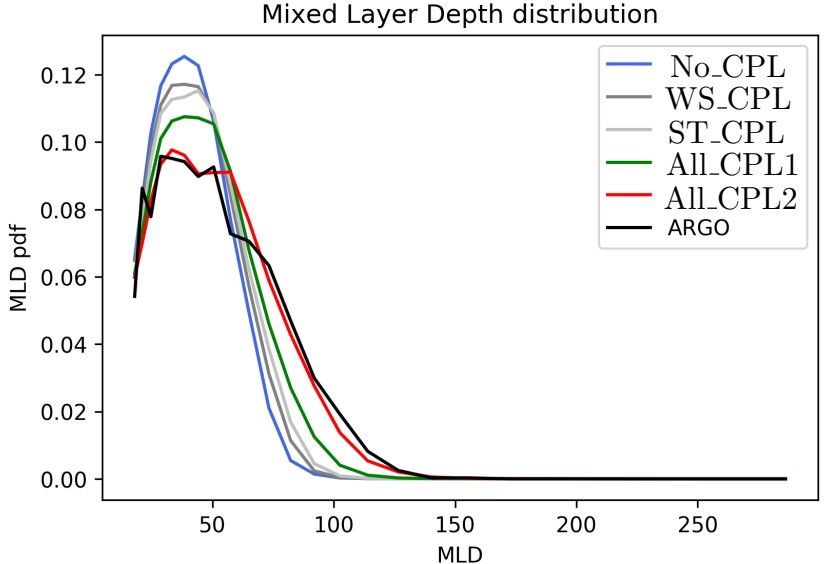

**Figure 10.** Mixed Layer Depth probability density function in the southern hemisphere during summer months. The details of each experiment can be found in Tab. 2.

induced by the waves coupling (Fig. 13,b). Similarly we can also observe a warming in the tropical and equatorial regions (Fig. 13,b) corresponding to the cold bias seen in Fig. 13 (panel a). In the southern extra-tropical region, a summer cooling is observed. It is induced by the wave coupling whereas Fig. 13 (panel a) shows a slight warm bias. During winter we can observe north of 60 S a warming in Fig. 13 (panel b) which again partially corresponds to a cold bias in Fig. 13 (panel a).

**4.2.5   Surface current and Kinetic Energy**

The last aspect of our solutions we would like to evaluate is the impact of the surface waves on surface currents and kinetic energy (KE). To do so, we show in Fig. 14 time series of the spatially averaged surface kinetic energy for both hemispheres. Whatever the hemisphere there is a net decrease of surface KE (up to 20% in the south) when a coupling with the waves is included. This decrease of surface kinetic energy reflects a decrease of surface currents magnitude. Indeed, as detailed in Fig.

15 which represents the vertical profile of the horizontal components of the current in the oceanic surface boundary layer, the coupling with waves decreases both the surface currents magnitude and the shear. While currents from the WS_CPL are increased due to increased wind stress, the Stokes Coriolis force when included in momentum equations leads to a decrease of velocities in the whole boundary layer as previously shown by Rascle et al. (2008) (orange lines in Fig. 15). Inclusion of the vertical mixing due to waves and Langmuir circulation attenuates the currents in the surface layer, resulting in further reduced

surface currents and stronger currents at the bottom of the boundary layer (purple lines in Fig. 15). This concludes our checking of the proper functioning of the coupling with waves as described in the present paper.



a)

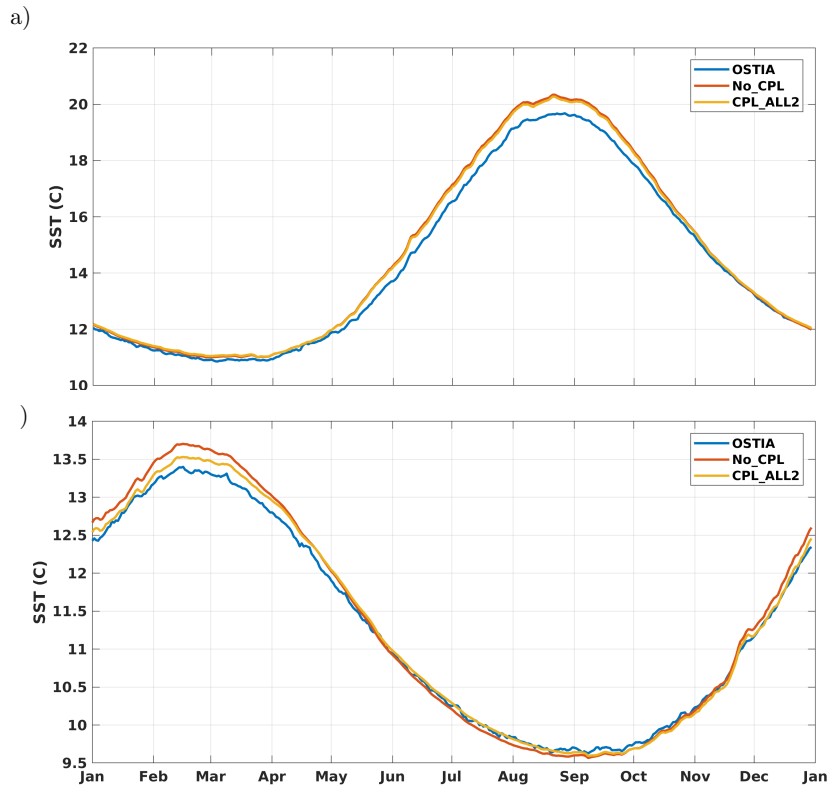

**Figure 11.** Time series of the spatially averaged Sea Surface Temperature ($^{o}$C); (a): North Hemisphere and (b): South Hemisphere

## 5 Conclusions

In this paper we have described the implementation of an online coupling between the oceanic model NEMO and the wave model WW3. The impact of such coupling on the model solutions has been assessed from the oceanic point of view for a

global configuration. In particular, the following steps to set up the coupled model have been discussed in details (*i*) inclusion of all wave-induced terms in NEMO primitive equations, only neglecting the terms relevant for the surf zone which is outside the scope of the NEMO community, (*ii*) modification of the subgrid scales vertical physics (including the bulk formulation) to include wave effects and a parameterization of Langmuir turbulence, (*iii*) development of a coupling interface based on the OASIS3-MCT software for the exchange of data between both models, and (*iv*) tests of our developments on a realistic

global configuration at $1/4°$ for the ocean coupled to a $1/2°$ resolution wave model. Compared to an ocean-only simulation, the coupling with a wave model (with a resolution twice coarser than the oceanic model) leads to an additional computational cost of about 20%.

Following McWilliams et al. (2004) and Ardhuin et al. (2008), in the weak vertical current shears limit, the wave-induced terms implemented in NEMO include the Stokes-Coriolis force, the vortex force, Stokes advection in tracer and continuity

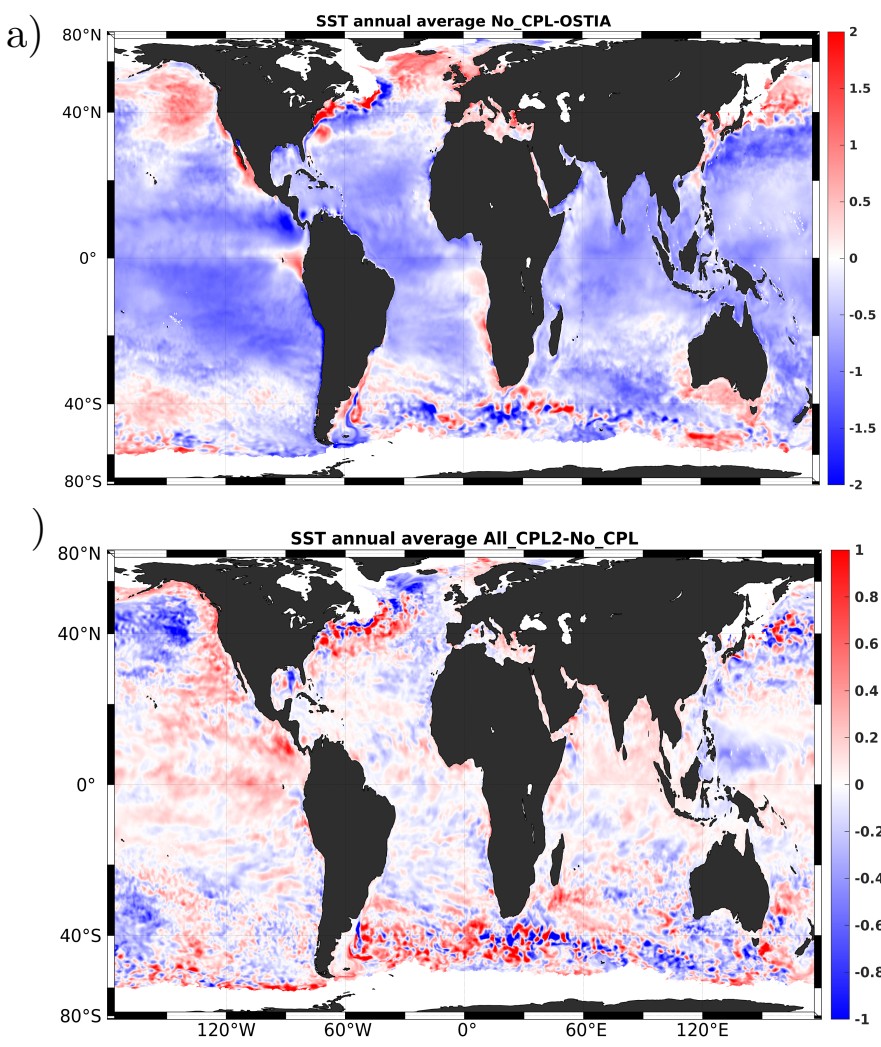

**Figure 12.** (a):Annual average of the differences between No_CPL and OSTIA sea surface temperatures ($^{o}$C) for year 2014 (positive when the model is warmer). (b):Annual average of the difference between All_CPL2 and No_CPL (positive when All_CPL2 is warmer)





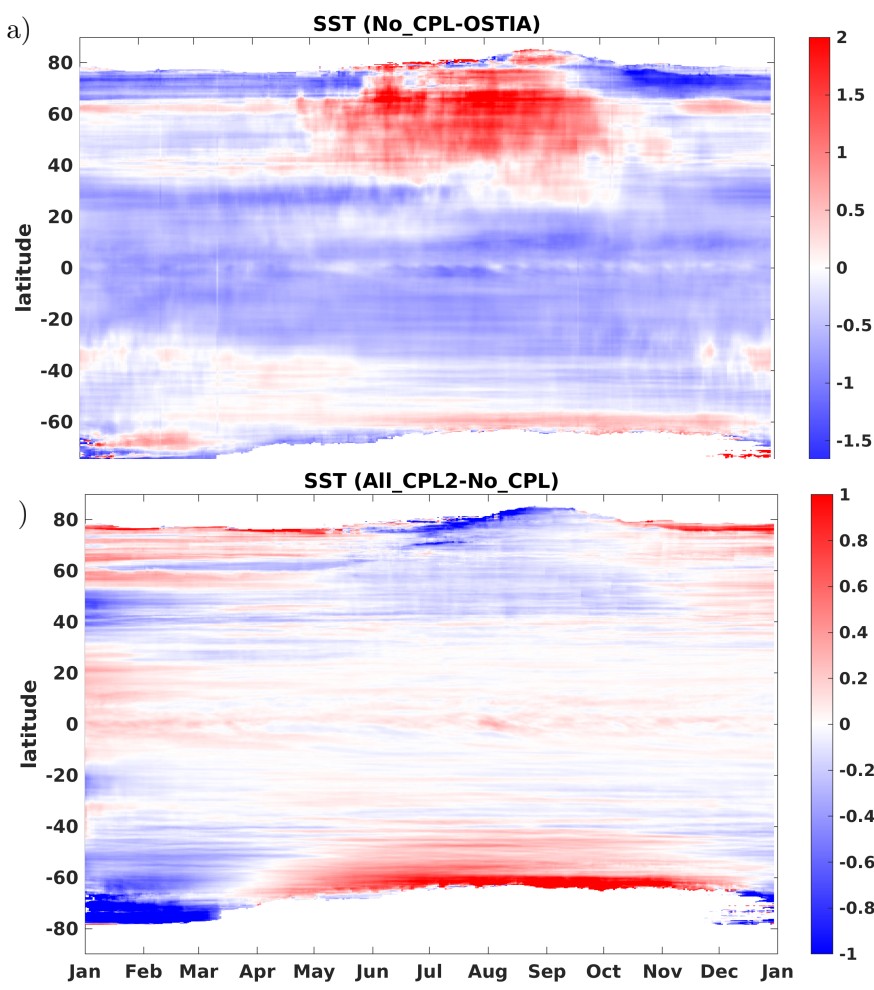

**Figure 13.** Hovmuller diagram of the longitudinally averaged sea surface temperature (°C) differences between (a): No_CPL and OSTIA and (b): between All_CPL2 and No_CPL.




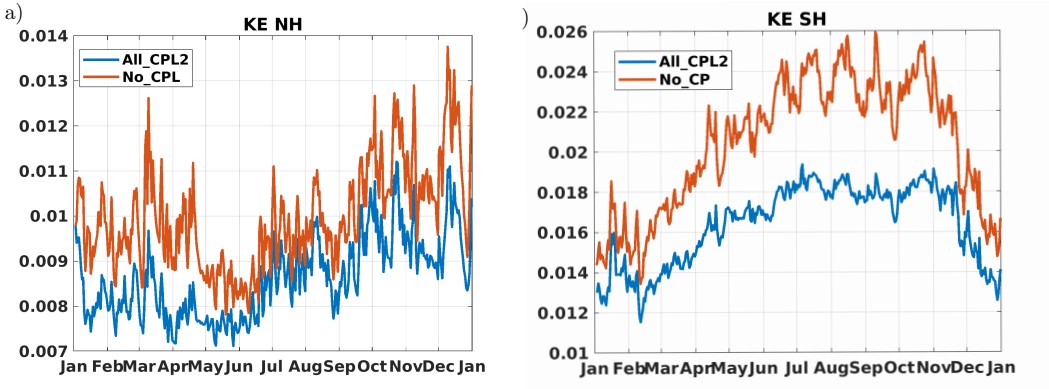

**Figure 14.** Time series of the spatially averaged surface kinetic energy ($m^2\,s^{-2}$) for (a): the Northern hemisphere and (b) the Southern hemisphere

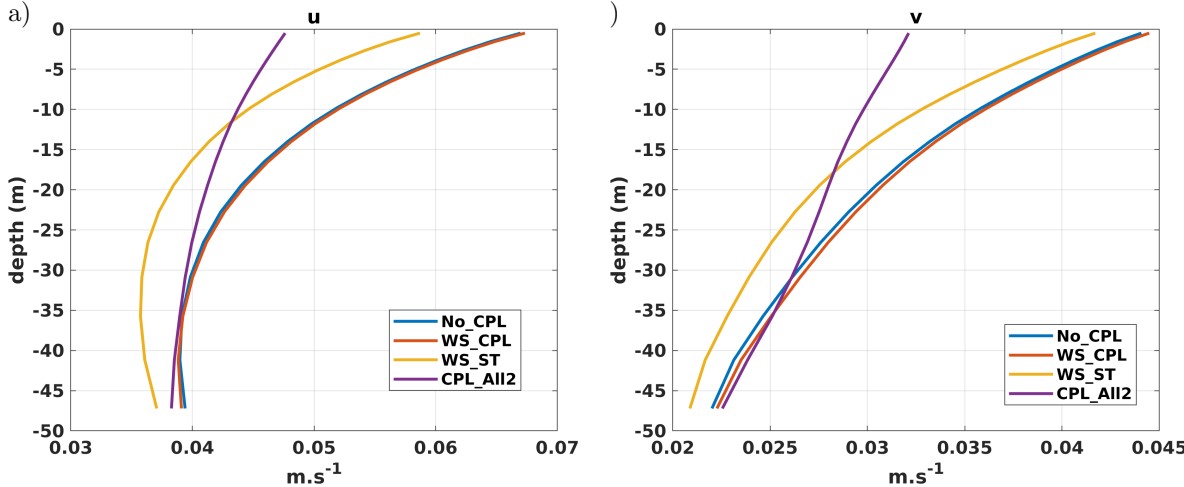

**Figure 15.** Zonally averaged zonal (a) and meridional (b) currents ($m\,s^{-1}$) between $60\,S$ and $25\,S$ as a function of depth (m) for the simulations described in Tab. 2.





equations as well as a wave-induced surface pressure term. The prognostic equation for TKE also includes an additional forcing term associated with the Stokes drift vertical shear as well as various modifications of its boundary condition described in Sec. 2.3.

The development of a coupling infrastructure based on OASIS3-MCT has several advantages as it allows for an efficient data exchange (including the treatment of non-conformities between the computational grids) but also for versatility in the

inclusion of a wave model in existing ocean-atmosphere or ocean-only models. At a practical level, the OASIS interface we have implemented in NEMO is similar to other interfaces (e.g. toward atmospheric models) existing in the code which is important for maintenance and for further developments. It paves the way for a seamless and more systematic inclusion of the coupling with waves for NEMO users.

Unlike most previous studies of wave-ocean coupling using NEMO, we have shown that satisfactory results can be obtained

from the TKE vertical turbulent closure scheme without activating the *ad hoc* parameterization for the mixing induced by near-inertial waves, surface waves and swells (known as ETAU parameterization). This parameterization which amounts to empirically propagate the surface TKE at depth using a prescribed shape function is a pragmatic way to cure the shallow mixed layer depths in the southern ocean found in simulations ignoring wave effects. Previous studies of wave-ocean coupling by Breivik et al. (2015), Alari et al. (2016) or Staneva et al. (2017) have been using the ETAU parameterization in their

setup. However, as suggested by Breivik et al. (2015), we can speculate that such parameterization could mask the impact of the wave coupling even though it turned out to be necessary to obtain realistic mixed layer depths. We believe that our modifications of the standard NEMO 1-equation TKE scheme described in Sec. 2.3 are more physically justifiable than the ETAU parameterization and require much less parameter tuning.

The numerical experiments based on the ORCA25 configuration discussed in Sec. 4.2 were meant to check that our de-

velopments were having the expected impact on numerical solutions. First, we confirmed that using the Charnock parameter computed in the wave model instead of a constant value globally increases the wind-stress magnitude, particularly at mid and high latitudes whereas accounting for the portion of the wind-stress consumed by the waves has a small impact (in our experiments it leads to a maximum of 2% decrease of the wind-stress). Second, using the mixed layer depth as an indicator to assess the amount of vertical mixing, the modifications brought to the NEMO turbulence scheme (i.e the new boundary

condition for TKE and for the mixing length, the addition of the Stokes shear in the TKE equation, and the modified Axell (2002) parameterization for Langmuir cells) lead to an important extra mixing contributing to a deepening of the surface mixed layer particularly in the southern hemisphere. When compared to ARGO data it shows a significant improvement during the summer, while during the winter the extra mixing induced by waves deepens the already too deep mixed layer. Note that the Fox-Kemper et al. (2008) parameterization to account for the restratification induced by mixed layer instabilities (Boccaletti

et al., 2007; Couvelard et al., 2015) during the winter was not used in our experiments. This parameterization induces even more shallow summer mixed layer depths. As far as the northern hemisphere is concerned, coupled results show an improvement when compared to ARGO for winter with a deepening of the mixed layer while in summer results are similar to the uncoupled case. Since the comparison with ARGO data can be tricky due to the scarcity of the data, we looked at the results in terms of mixed layer depths (MLD) probability density functions. This allowed to highlight the significant improvement in



MLD distribution when coupling with the waves. Furthermore, we noticed that all components of the ocean-wave coupling act to deepen the mixed layer and therefore have a cumulative effect. However the main contributor is the modified Langmuir cell parameterization of Axell (2002) which is consistent with recent results obtained by Reichl et al. (2016) and Ali et al. (2019) using a KPP closure scheme.

Since the magnitude of the vertical mixing is increased by the coupling with waves we expect an impact on sea surface
temperature and currents. Indeed, the summer deepening of the mixed layer in the southern hemisphere leads to colder sea surface temperatures resulting in a better agreement with OSTIA SST analysis. More generally, although the global SST biases are not totally compensated, they tend to be reduced when considering the effect of waves (see Sec. 4.2.4). The currents in the oceanic surface boundary layer are reduced by the Stokes Coriolis force (which counteracts the Ekman current, Rascle et al., 2008). They are also affected by the increased vertical mixing which tends to reduce the surface currents (and thus the surface
kinetic energy) and strengthen the currents at the base of the surface boundary layer. The reduction of surface kinetic energy due to the wave-ocean coupling in the global $1/4°$ resolution configuration is of the same order of magnitude as the reduction observed when accounting for surface currents in the computation of the wind stress in a coupled ocean-atmosphere model (e.g. Renault et al., 2016). A fully coupled ocean-wave-atmosphere model would thus be necessary to properly disentangle the different contributions at play impacting the oceanic surface kinetic energy. Even if additional diagnostics on various
configurations at different resolutions are still needed to exhaustively evaluate the impact of each component of the ocean wave coupling, the results presented in the paper confirm the robustness of our developments and our implementation will serve as a starting point for the inclusion of wave-currents interactions in the forthcoming NEMO official release. We can speculate that the ocean-waves coupled ORCA025 configuration might become a standard component of future Coupled Model Intercomparison Project (CMIP) exercises. We already mentioned as a perspective the addition of a coupling with an interactive
atmospheric boundary layer either via a full atmospheric model or a simplified boundary layer model (e.g. Lemarié et al., 2019). Furthermore, the gain of an online 2-way coupling compared to a 1-way coupling on the oceanic as well as on the wave solution must be investigated in the future. Indeed, the improvements of the quality of surface waves simulations associated to a coupling with large-scale oceanic currents are well documented particularly in the Agulhas current (Irvine and Tilley, 1988) and in the Gulf Stream (Mapp et al., 1985). Ardhuin et al. (2017a) have also shown a strong impact of small-scale currents
(10-100km) on wave height variability at the same scales. We can therefore expect improvements for both wave and ocean forecasts when the coupling is implemented in an operational context.

*Code and data availability.* The changes to the NEMO code have been made on the standard NEMO code (nemo_v3_6_STABLE). The code can be downloaded from the NEMO website (http://www.nemo-ocean.eu/, last access: 11 July 2019). The NEMO code modified to include wave-ocean coupling terms and the OASIS interface is available in the zenodo archive (https://doi.org/10.5281/zenodo.3331463,
Couvelard (2019)). The WW3 code version 6.02 has been used without further modifications and can be downloaded from the NOAA github repository (https://github.com/NOAA-EMC/WW3, last access: 11 July 2019). Our modifications of the OASIS interface in the WW3 code have already been integrated in the official release. The OASIS3_MCT code is also freely available (https://portal.enes.org/oasis/, last access:





11 July 2019). The exact versions of the WW3 and OASIS3_MCT codes that were used have also been made available in the zenodo archive
(https://doi.org/10.5281/zenodo.3331463, Couvelard (2019)) The initial and forcing data for both the oceanic and wave model, analysis
scripts, namelists and data used to produce the figures are also available in the zenodo archive.

## Appendix A: Flux-form wave-averaged momentum equations

In this appendix we describe the necessary changes when a flux formulation for advective terms in the momentum equations is preferred to the vector invariant form presented in (7) and (8). For simplicity, we consider just the $i$-component in horizontal curvilinear coordinates and a $z$-coordinate in the vertical (results will be extended to the $j$-component and to generalized

vertical coordinate). Consistently with the notations of Madec (2012), $e_1$ and $e_2$ are the horizontal scale factors. We note $A_v^u$ the extra term needed to guarantee the equivalence between the flux formulation and the vector-invariant form. $A_v^u$ is defined such that

$$\nabla \cdot (\mathbf{u}^s u) + A_v^u = -\zeta v^s + \frac{w^s}{e_3} \partial_k u.$$

Since $\nabla \cdot \mathbf{u}^s = 0$ we have $\nabla \cdot (\mathbf{u}^s u) = \mathbf{u}^s \cdot \nabla u$, and thus

$$
\begin{aligned}
\quad e_1 e_2 A_v^u &= -v^s [\partial_i(e_2 v) - \partial_j(e_1 u)] + \frac{e_1 e_2}{e_3} w^s \partial_k u - \left[ e_2 u^s \partial_i u + e_1 v^s \partial_j u + \frac{e_1 e_2}{e_3} w^s \partial_k u \right] \\
&= -v^s [v \partial_i e_2 - u \partial_j e_1 + e_2 \partial_i v - e_1 \partial_j u] - e_2 u^s \partial_i u - e_1 v^s \partial_j u
\end{aligned}
$$

hence

$$A_v^u = \underbrace{-\frac{v^s}{e_1 e_2}(v \partial_i e_2 - u \partial_j e_1)}_{\text{Metric term on Stokes drift}} \underbrace{-\left( \frac{u^s}{e_1}\partial_i u + \frac{v^s}{e_1}\partial_i v \right)}_{\text{Additional term}}$$

Same computation for the $j$-component leads to the following equations in generalized vertical coordinates

$$
\begin{aligned}
\quad \frac{1}{e_3}\partial_t(e_3 u) &= -\frac{1}{e_1 e_2}[\partial_i(e_2(u+\widetilde{u}^s)u) + \partial_j(e_1(v+\widetilde{v}^s)u)] + \frac{1}{e_3}\partial_k((\omega+\widetilde{\omega}^s)u) + \left[f + \frac{1}{e_1 e_2}(v\partial_i e_2 - u\partial_j e_1)\right](v+\widetilde{v}^s) + \frac{\widetilde{u}^s}{e_1}\partial_i u + \frac{\widetilde{v}^s}{e_1}\partial_i v \\
&\quad - \frac{1}{\rho_0 e_1}\partial_i(p_s + \widetilde{p}^J) - \frac{1}{\rho_0 e_1}\left(\partial_i p_h - \frac{(\partial_k p_h)(\partial_i z)}{e_3}\right) + \frac{1}{e_3}\partial_k\langle u'w'\rangle + F^u + \widetilde{F}^u \\
\frac{1}{e_3}\partial_t(e_3 v) &= -\frac{1}{e_1 e_2}[\partial_i(e_2(u+\widetilde{u}^s)v) + \partial_j(e_1(v+\widetilde{v}^s)v)] + \frac{1}{e_3}\partial_k((\omega+\widetilde{\omega}^s)v) - \left[f + \frac{1}{e_1 e_2}(v\partial_i e_2 - u\partial_j e_1)\right](u+\widetilde{u}^s) + \frac{\widetilde{u}^s}{e_2}\partial_j u + \frac{\widetilde{v}^s}{e_2}\partial_j v \\
&\quad - \frac{1}{\rho_0 e_2}\partial_j(p_s + \widetilde{p}^J) - \frac{1}{\rho_0 e_2}\left(\partial_j p_h - \frac{(\partial_k p_h)(\partial_j z)}{e_3}\right) + \frac{1}{e_3}\partial_k\langle u'w'\rangle + F^u + \widetilde{F}^u
\end{aligned}
$$

## Appendix B: Sensitivity to the $c_{\text{LC}}$ parameter from single-column experiments

Single column experiments based on Noh et al. (2016) have been performed to study the behavior of the NEMO vertical closure with the Langmuir cells parameterization of Axell (2002). In the Noh et al. (2016) experiments the initial condition is given by

$$u(z,t) = v(z,t) = 0, \qquad \theta(z,t) = \min\left\{ T_0 - N_0^2 \frac{(z-5.)}{\alpha g}, T_0 \right\}$$





with $\alpha$ the thermal expansion coefficient in the equation of state defined as $\rho = -\alpha\rho_0(T - T_0)$ with $\rho_0 = 1024\,\mathrm{kg\,m^{-3}}$. A

zonal wind is imposed with $u_\star = 0.02\,\mathrm{m\,s^{-1}}$ and the Stokes drift is given by

$$\mathbf{u}_s = (u_s, 0), \qquad u_s = \left(\frac{2\pi a}{\lambda}\right)^2 \sqrt{\frac{g\lambda}{2\pi}} e^{-4\pi z/\lambda}$$

The various parameter values are

$$f_{\mathrm{cor}} = 10^{-4}\,\mathrm{s^{-1}}, \qquad h_{\max} = 120\,\mathrm{m}, \qquad T_0 = 16\,^0\mathrm{C}, \qquad N_0^2 = 10^{-5}\,\mathrm{s^{-2}}$$

with 96 vertical levels for the discretization and 16 hours simulations. We only consider the case with $a = 1\,\mathrm{m}$ and $\lambda = 40\,\mathrm{m}$

which gives a turbulent Langmuir number of $\mathrm{La}_t \approx 0.32$. Numerical results are shown in Fig. B1 (upper panels) and are

consistent with the results of Noh et al. (2016) with a deepening of the oceanic mixing length of about $10\,\mathrm{m}$ when Langmuir

turbulence is accounted for. For $C_{\mathrm{LC}} = 0.15$ in the Axell (2002) parameterization, the deepening is too weak while for $C_{\mathrm{LC}} =$

$0.3$ it is closer to LES results. Note that for those experiments, the value of $d_{\mathrm{LC}}$ is almost identical to the mixed layer depth. Fig.

B1 (lower panels) illustrates the fact that for a stronger stratification (i.e. with $N_0^2 = 2 \times 10^{-4}\,\mathrm{s^{-2}}$ instead of $N_0^2 = 10^{-5}\,\mathrm{s^{-2}}$)

the effect of Langmuir turbulence on mixed-layer depth is negligible. Indeed in this case Langmuir cells do not provide enough

mixing to erode the stratification.

*Author contributions.* Xavier Couvelard has prepared and carried out all the numerical experiments, has investigated the results, and wrote

the paper with the help of all the coauthors. Gurvan Madec, Florian Lemarié, and Rachid Benshila have made the changes in the NEMO

code to include the wave-ocean interactions. Guillaume Samson helped to prepare the necessary datasets for the numerical experiments

and analyse the model outputs. Fabrice Ardhuin and Jean-Luc Redelsperger helped to investigate the results and to formalize the necessary

wave-induced terms both in the primitive equations and in the TKE closure.

*Competing interests.* The authors declare that they have no conflict of interest.

*Acknowledgements.* Xavier Couvelard, Florian Lemarié and Jean-Luc Redelsperger acknowledge the support by Mercator-Ocean and the

Copernicus Marine Environment Monitoring Service (CMEMS) through contract 22-GLO-HR - Lot 2 (High-resolution ocean, waves, at-

mosphere interaction). Numerical simulations were performed on Ifremer HPC facilities DATARMOR of "Pôle de Calcul Intensif pour la

Mer" (PCIM) (http://www.ifremer.fr/pcim). Mixed Layer Depth Data were graciously provided by Clément de Boyer Montégut, and SST

data were downloaded from the CMEMS catalogue. The Authors also greatfully thanks Claude Talandier for its help with NEMO, Mickaël

Accensi for its help with WW3 and Eric Maisonnave and Laure Coquart for their help with OASIS3_MCT.



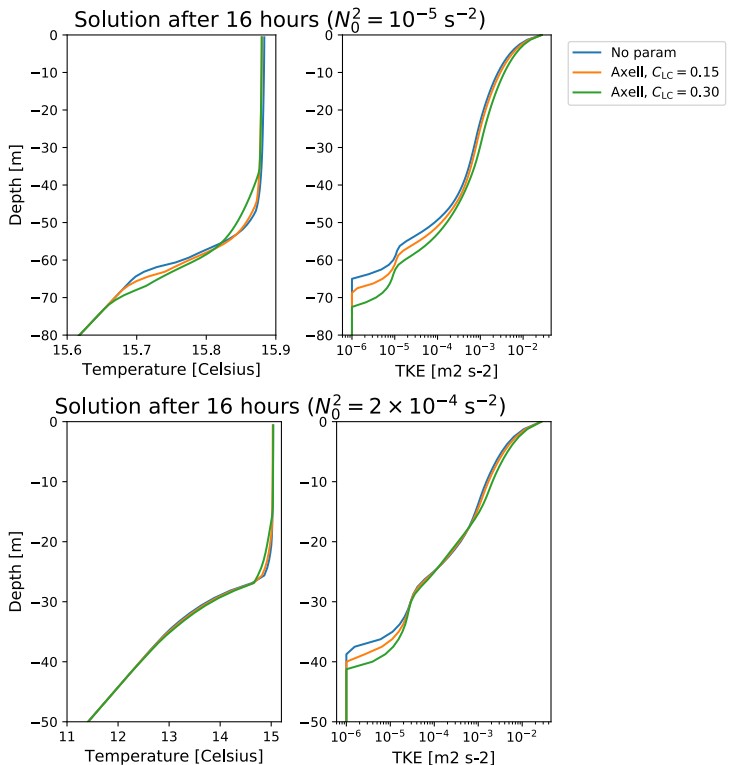

**Figure B1.** Solution obtained for the Noh et al. (2016) single column experiment after 16 hours for different parameter value in the Axell (2002) Langmuir cell parameterization in the case $N_0^2 = 10^{-5}\,\mathrm{s}^{-2}$ (upper panels) and $N_0^2 = 2 \times 10^{-4}\,\mathrm{s}^{-2}$ (lower panels).

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
