# Peer review of "Development of a 2-way coupled ocean-wave model: assessment on a global NEMO(v3.6)-WW3(v6.02) coupled configuration"

_Geoscientific Model Development, 2019_

## Referee Comment (RC1) · Oyvind Breivik (Referee) · 4 Oct 2019

This paper describes a two-year experiment with a coupled WW3-NEMO setup. The experiment builds on earlier experiments by Breivik et al (2015) and others who investigated the impact of waves on the mixed layer. The paper is well written and clear.

First, the change from a Dirichlet to a Neumann condition for the TKE flux should be discussed in more detail. It is not clear to me that comparing against an uncoupled run with a Dirichlet conditon is clean. A separate experiment should be run where the uncoupled model ingests a flux in the Neumann form, or alternatively a coupled run where the Dirichlet condition is used to communicate the TKE flux from WW3.

[Figure]

The integration period is rather short. I think the authors should investigate whether there is sufficient convergence after just two years.

The Langmuir experiment is very interesting as it promises a way forward from the ETAU hack. I would like to see a quantification of how much changing from parameterized Stokes drift (1.6% of the wind speed) to a Stokes drift taken from WW3 gives you. I suspect the most important thing you've done is to chance the factor from 0.15 to 0.30.

Further on the Langmuir experiment, you don't seem to improve the

The Stokes drift discussion is interesting. I suggest you read the appendix of Li et al (2017) where there is a description of the finite volume form of the profile by Breivik et al (2016). Also, the recent paper by Wu et al (2019) discusses the combined impact (quite small!) of the Coriolis-Stokes force and the Stokes drift on tracer advection.

Finally, a quantitative assessment of the relative impact of the various wave-induced processes is needed in order to give the reader an idea of their importance. This applies to the description in Sec 4.2.3 as mentioned below.

Cost: You have run WW3 on half the resolution of NEMO at 20% added cost. Have you considered the added benefit of running the models on similar resolution? I presume this would cost more than twice the standalone NEMO run, so I sympathize with your decision, though.

All told, I would say that after major revision (rerunning the experiments with Dirichlet or Neumann to make a clean comparison) and assessment of the relative importance of the wave-induced effects, this paper should be acceptable for publication in GMD.

Detailed comments:

Fig 2 is a mess. Please explain in detail what is shown in the different panels and refer to those panels in the text. The figure headings are illegible. I am also surprised by the huge difference in average wave height and would like to see a more in-depth

discussion of why this is so.

4.2.2 It is interesting that you have rewritten the Dirichlet conditon to a Neumann condition for the TKE flux. However, I think you should also investigate how this affects the results as you compare against an uncoupled run with a Dirichlet condition.

4.2.3 The impact on MLD and SST does not separate between Langmuir, TKE flux and stress. This needs to be done.

References:

Wu, L, J Staneva, O Breivik, A Rutgersson, A G Nurser, E Clementi, G Madec (2019). Wave effects on coastal upwelling and water level, Ocean Model, 140, p 101405, doi:10.1016/j.ocemod.2019.101405

————————————————

---

## Referee Comment (RC2) · Oyvind Breivik (Referee) · 4 Oct 2019

Sorry, a comment was left dangling. I was meant to say about the Langmuir mixing that you don't seem to improve the MLD much, but this is part of the general comment I was making that you need to separate the impact of the various processes.

---

## Referee Comment (RC3) · A.J. George Nurser (Referee) · 14 Nov 2019

**A review of GMD-20180189: Development of a 2-way coupled ocean-wave model: assessment on a global NEMO(v3.6)-WW3(v6.02) coupled configuration**

by Xavier Couvelard, Florian Lemarié, Guillaume Samson, Jean-Luc Redelsperger, Fabrice Ardhuin, Rachid Benshila, and Gurvan Madec

Firstly, I need to apologize for the very late review.

**Summary**

This MS describes the implementation of a 2-way coupled wave–ocean model, involving v3.6 NEMO and Wavewatch 3 v6.03. This model aims to include most of the wave impacts on the ocean. It includes

1. 2-way coupling of the ocean and wave models using the OASIS3-MCT coupler. Surface ocean velocities are passed to the wave model, and various wave properties, including surface and depth-integrated Stokes drift, energy dissipation rate, wave-ocean stress and Charnock parameter are passed to the ocean model.

2. Calculation of the wind-stress exerted by the atmosphere ($\tau^{\text{atm}}$) using a value of the Charnock parameter output from the wave model, which is itself forced by a stress $\tau^{\text{atm}}_{\text{WW3}}$ calculated from the 10-m winds using an approximate, constant (independent of atmospheric stability or Charnock number) drag coefficient. The surface-stress that drives the ocean is then $\tau^{\text{atm}} - \tau^{\text{atm}}_{\text{WW3}} + \tau^{\text{oce}}_{\text{WW3}}$, where $\tau^{\text{oce}}_{\text{WW3}}$ is the rate of momentum transfer from the waves to the ocean by breaking waves.

3. Wave-influence terms added to the momentum and tracer advection equations that are linked to the Stokes drift. The vortex-force term is included in the momentum equation, as well as the Stokes-Coriolis force included in previous implementations.

4. Use of the wave energy dissipation rate $\Phi_{\text{oc}}$ as a flux (Neumann) surface boundary condition for the TKE that is prognosed by the TKE model that is used here to specify vertical eddy diffusivities and viscosities.

5. A revised version of Axell's (2002) parameterization of Langmuir turbulence for the TKE model, with a doubling of the TKE source associated with vertical Langmuir turbulence imposed within an upper Langmuir-influenced part of the surface boundary layer.

The effects of including these wave impacts on the ocean are briefly examined through four 2-year simulations that introduce subsets of the above enhancements (1)–(5), involving v3.6 NEMO run at 1/4° resolution (ORCA025) coupled with WW3 run at 1/2° resolution. With the full model with all 5 additions, mixed-layer depths are found to be improved in the Southern Ocean, particularly in southern summer, when the standard model gives summer mixed-layers that are too shallow.

**Major Comments**

The authors have done a lot of work here in producing a coupled version of NEMO with WW3. The explanation of the extra terms added to NEMO is full, very much in the spirit of a GMD contribution, and it is good to see that the code is indeed publicly available. However, shortcuts have been taken e.g. the use of a neutral drag coefficient independent of Charnock number to estimate the atmospheric stress transferred into the waves, while the total atmospheric stress is separately calculated and depends on Charnock number and atmospheric stability.

The testing of the modifications with 2-year runs is rather cursory, but I guess that the intention of that short testing period is more to check that the code is basically OK rather than to optimize the parameterizations. However, there really should be a test run that includes the changes to the TKE model coming from the flux condition using wave-dissipation energy [(4) above] but not including the Langmuir cell parameterization.

The paper generally seems a bit rushed, and the English while being perfectly readable, is not great; there are many extra s's where there should be none, etc.

**Recommendation**

This MS should be accepted, subject to minor revisions.

**Detailed Comments**

**p2, l39** Should refer to Lu et al. (2019).

**p3, Eqs. (1)–(4)** Various $w$ should be $\omega$.

**p3, Eqs.(1)–(4)** Please define $p_h$ and $p_s$.

**p3, l78** $\tau^{\text{oce}}$ is not strictly the wind stress; it is that part of the stress that drives the ocean rather than developing the wave field.

**p3, l78** Please explain "the dynamic boundary condition imposing the continuity of pressure at the air-sea interface"

**p3, l80** $\omega(z = -H) = 0$. This is only true for terrain following coordinates, not for a generalized coordinate.

**p4, Eqs.(7)–(8)** Please define $p^J$ and $p^{\text{FV}}$. I assume $p^J$ is the $J$ that is only significant in shallow water, defined in eq. (20) of Bennis et al., (2011). If so, then presumably $p^{\text{FV}}$ represents the term $\frac{1}{2}[(\mathbf{u} + \mathbf{u}^s)^2 - \mathbf{u}^2]$ found e.g. on the RHS of Eq. (2) of Suzuki and Fox-Kemper (2016). This term would seem to scale with the vortex force term. Can the authors justify its neglect?

**p4, Eq.(12)** The "wave pressure vector" seems a little odd. It would be more natural to make $\mathcal{W}_{\text{Prs}}$ the column vector of x- y- and vertical gradients, especially given that in p5, l118 you refer to "the additional wave- induced barotropic forcing terms corresponding to the vertical integral of the ... $\mathcal{W}_{\text{Prs}}$"

**p5, Eq.(13)** How do you decompose baroclinic and barotropic contributions to bottom drag when using non-linear bottom drag, as you do in these experiments (p13, l342)?

**p6, Figure 1** You seem to have evaluated the primitive $\mathcal{I}_{\text{B14}}$, as you plot it out here. Why is it not written out explicitly like $\mathcal{I}_{\text{B16}}$, which is set out at the bottom of p5?

**p6, l140–141** This is a nice point. But note on l141 that "summed to $\omega$" should summed with $\omega$. More importantly, please briefly explain how $\omega + \omega^s$ is set; Eq. (10) looks more like a prognostic equation for $e_3$ than a diagnostic equation for $\omega + \omega^s$.

**p9, l211** Should be $|\mathbf{u}^s_{\text{LC}}| \propto \sqrt{(|\tau|)}$

**p11, l272–73** "most of the momentum flux going into the waves is quickly transferred to the water column through wave breaking (we call this fraction $\tau^{\text{oce}}$". Do you mean $\tau^{\text{oce}}_{\text{WW3}}$?

**p11, l276** The Charnock number is used to give the surface roughness that presumably in fact allows wind to drive waves. Why is the WW3 model then not forced by a drag coefficient that includes this effect? Also, why is the stress that drives the WW3 waves not stability-dependent?

**p11, Eq. (21)** This equation does seem to ensure that momentum is conserved, although I guess $\tau^{\text{atm}} - \tau^{\text{atm}}_{\text{WW3}}$ may be much bigger than it should be.

**p11, l289–291** I understood that the situation is not that clear, especially for eddy-resolving models, and that some consideration *does* need to be paid to the ocean current when calculating wind stress. E.g. the last sentence of Renault et al. (2018) states: " A simulation without current feedback—by overestimating the eddy amplitude, lifetime, and spatial range.."

**p13** Much of the first para seems to describe the wave model rather than its specific setup, so might fit in better into section 3.1.

**p13, , l337–8** "The numerical options are the one commonly chosen by the Drakkar group". This is a bit confusing; please indicate which of the options described here are the Drakkar options, and whether there are other option choices described in the Drakkar website that are not described here.

**p13, , l342** How does the lateral diffusivity vary away from the equator?

**p14, l348–351** More specific details and/or references are required here. Is it only solar forcing that is given a diurnal cycle? A reference is required for the data correction to ensure consistency.

**p14, section 4.1.3** There should be a test experiment with ST_CPL + changes to the TKE scheme but no Langmuir parametarization, to see whether the new TKE boundary condition makes any difference.

**p14, section 4.1.3** Please specify the initial conditions. Is it a spun up run of some standard NEMO setup? If so, give details.

**p15, Figure 2** Various random missing letters on panel titles.

**p16–p17, section 4.2.2** Given the amount of space devoted to the extra TKE injection (& 2 figures!), it really does seem strange that no run with ST_CPL + changes to the TKE scheme but no Langmuir parameterization has been presented.

**p17–p19, section 4.2.3** Give reference for ARGO MLD climatology, and specify the MLD criteria used in model and climatology.

**p17–p19, section 4.2.3** Maps of discrepancies of MLD from ARGO, and zonal-average MLDs would be more convincing than the MLD pdfs.

**p20, l444–445** "an increased heat content during winter leading to higher SST during summer." Is this the wrong way round?

**p29, appendix B** It is not easy to see which of these solutions is best. On p9, l25–16, you write " Based on single-column experiments detailed in App. B, we find that parameter values in the range 0.15 − 0.3 provide satisfactory results compared to LES simulations" Where are these LES simulation results?

—George Nurser

---

## Author Comment (AC1) · 17 Feb 2020

**Review from George Nurser**

First, the authors would like to sincerely thank the reviewers for their careful reading of the paper and their valuable comments to the manuscript and helpful suggestions. We further clarified several issues raised during the review process. Please find attached our revised paper and below a summary of how we responded to the comments. Our comments are reported in color in the text below.

[Figure]

Major Comments

- The authors have done a lot of work here in producing a coupled version of NEMO with WW3. The explanation of the extra terms added to NEMO is full, very much in the spirit of a GMD contribution, and it is good to see that the code is indeed publicly available.
  Thanks for this encouraging comment, the code is indeed publicly available and the developments are now in the process of being incorporated in the official NEMO release within the H2020 IMMERSE project.

- However, shortcuts have been taken e.g. the use of a neutral drag coefficient independent of Charnock number to estimate the atmospheric stress transferred into the waves, while the total atmospheric stress is separately calculated and depends on Charnock number and atmospheric stability.
  From your remark, it seems that our description of how the surface wind-stress is computed was not clear enough. The computation of the wind-stress in the wave model and in the oceanic model are both function of the Charnock parameter computed by the wave model. As mentionned in Sec. 3.2, in the wave model the general formula used is $\boldsymbol{\tau}_{\text{ww3}} = \rho_a C_{\text{DN}} \|\mathbf{u}_{10}^{\text{atm}}\| \mathbf{u}_{10}^{\text{atm}}$ where $C_{\text{DN}} = \left( \dfrac{\kappa}{\ln\left(\frac{z}{z_0}\right)} \right)^2$ where $z_0$ depends on the Charnock parameter. The only difference between $\boldsymbol{\tau}_{\text{ww3}}$ and the stress computed in the oceanic model is that the latter accounts for atmospheric stability. Note that what you call a "shortcut" is what is actually done in all coupled ocean-wave models as none of them guarantees energetic consistency (this point is often swept under the carpet in publications). Let us mention that :

  – Wave models in "forced mode" do not have any information on atmospheric temperature/humidity or SST which explains why they neglect atmospheric

stability in the wind-stress computation.

– The solution of wave models is very sensitive to wind-stress and our wave configuration has been designed and validated in forced mode with neutral drag coefficient. We tried to run WW3 with the same bulk formulation as NEMO but the quality of the wave solution was drastically deteriorated doing so.

We tried to further clarified those aspects in the revised manuscript.

• The testing of the modifications with 2-year runs is rather cursory, but I guess that the intention of that short testing period is more to check that the code is basically OK rather than to optimize the parameterizations. However, there really should be a test run that includes the changes to the TKE model coming from the flux condition using wave dissipation energy [(4) above] but not including the Langmuir cell parameterization.
Following your suggestion we have added a new case TKE_CPL in Tab. 2 which includes all ingredients but the Langmuir Cells parameterization. The results thus obtained are showed and discussed in Fig. 10 and Sec. 4.2.3.

• The paper generally seems a bit rushed, and the English while being perfectly readable, is not great; there are many extra s's where there should be none, etc. Sorry for that, we tried to correct as much as possible these issues.

Detailed Comments

• p2,l39 Should refer to Lu et al. (2019).
We believe that instead of Lu et al. (2019) it should be Wu et al. (2019) ? A reference to Wu et al. (2019) has been added, in particular they also compute the Stokes drift as a "layer-averaged Stokes drift profile" based on the Breivik et al. (2016) profile. It is really not clear from their paper but it seems that they

introduced in NEMO a Neumann (flux) boundary condition for the TKE equation. Thanks for pointing out this reference to us.

- p3,Eqs. (1)–(4) Various w should be $\omega$.
  It has been corrected in subgrid scale terms

- p3,Eqs.(1)–(4) Please define $p_h$ and $p_s$.
  Done

- p3,l78 $\tau^{\text{oce}}$ is not strictly the wind stress; it is that part of the stress that drives the ocean rather than developing the wave field.
  Indeed you are right, it has been changed.

- p3,l78 Please explain "the dynamic boundary condition imposing the continuity of pressure at the air-sea interface"
  We do not necessarily see what should be explained here. There must be no pressure jump at the air-sea interface, continuity of pressure translates into $p = p_{\text{atm}}$ at the interface with $p_{\text{atm}}$ the atmospheric pressure at sea surface.

- p3,l80 $\omega(z = -H) = 0$. This is only true for terrain following coordinates, not for a generalized coordinate.
  Since $\omega$ is the dia-surface velocity component, at the lower boundary the no-normal flow boundary condition should read $\omega_{\text{bot}} = 0$ which is equivalent to $\mathbf{u} \cdot \mathbf{n} = 0$. We do not understand the issue here, the continuity equation equation integrates starting from $\omega = 0$ even with geopotential coordinate.

- p4,Eqs.(7)–(8) Please define $p^{\text{J}}$ and $p^{\text{FV}}$. I assume $p^{\text{J}}$ is the J that is only significant in shallow water, defined in eq. (20) of Bennis et al., (2011). If so, then presumably $p^{\text{FV}}$ represents the term $\frac{1}{2}\left[(\mathbf{u} + \mathbf{u}^s)^2 - \mathbf{u}^2\right]$ found e.g. on the RHS of Eq. (2) of Suzuki and Fox-Kemper (2016). This term would seem to scale with the vortex force term. Can the authors justify its neglect?

Thanks for raising this issue. First we tried to clarify the notations in the paper and the way the additional wave related terms are introduced. In Suzuki and Fox-Kemper (2016) (SFK16) the Craik-Leibovich (CL) equations are used while our implementation relies on the more general wave-averaged primitive equations. In the CL equations the Bernoulli head term (let us note it $\mathcal{K}$) is defined as the kinetic energy increase due to the waves, i.e. $\mathcal{K} = \dfrac{\|\mathbf{u}^s + \mathbf{u}\|^2}{2} - \dfrac{\|\mathbf{u}\|^2}{2}$. In the wave-averaged equations, the form of the Bernoulli head is much more compli-cated (see eq (9.20) in McWilliams et al. 2004 or the $S^{\text{Shear}}$ term in Eq. (40) in Ardhuin et al. 2008) and it does not appear explicitly in our implementation be-cause of the general weak vertical shears in the wave-mixed layer. The effect of that term was also found to be much weaker than $S^J$ in shallow coastal environ-ments, except in the surf zone. It is also mentioned in SFK16 (their Eq (14)) that the contribution of the Stokes shear force should be retained but since this term results from the combination of the vertical component of the vortex force with the Bernoulli head, it does not appear explicitly in our derivation. Finally, compared to the previous version of the manuscript, additional terms related to the slope of the vertical coordinate have been added in Eqs. (7) and (8). Those terms are pieces of the vortex force which need to be taken into account with a generalized vertical coordinate. They seldom appear in the literature because most people present the wave-averaged equations in geopotential coordinate.

- p4,Eq.(12) The "wave pressure vector" seems a little odd. It would be more natu-ral to make $\mathcal{W}_{\text{prs}}$ the column vector of x- y- and vertical gradients, especially given that in p5, l118 you refer to "the additional wave-induced barotropic forcing terms corresponding to the vertical integral of the ... $\mathcal{W}_{\text{prs}}$"
  This has been reformulated by including the gradients in the $\mathcal{W}_{\text{prs}}$ term and re-moving the reference to the vertical integral of $\mathcal{W}_{\text{prs}}$ since after our simplifications this becomes a 2D horizontal field. The notations have been adapted and are now more consistent with Bennis et al. (2011) and Michaud et al. (2012) except

that the $S^J$ and $S^{\text{Shear}}$ terms are expressed directly in terms of pressure terms $\widetilde{p}^J$ and $\widetilde{p}^{\text{Shear}}$.

- p5,Eq.(13) How do you decompose baroclinic and barotropic contributions to bottom drag when using non-linear bottom drag, as you do in these experiments (p13, l342) ?
  The non-linear bottom drag in the baroclinic mode is computed in an implicit way as $(C_D\|\mathbf{u}_h\|)^{n-1}\,\mathbf{u}_h^{n+1}$. Because of the linearization this term is analogous to a linear bottom drag and thus easy to separate into a barotropic and a baroclinic contributions. See Sec. 10.4 in the version 4.0 of the NEMO documentation for more details.

- p6,Figure1 You seem to have evaluated the primitive $\mathcal{I}_{\text{B14}}$,as you plot it out here. Why is it not written out explicitly like $\mathcal{I}_{\text{B16}}$, which is set out at the bottom of p5?
  The primitive of $\mathcal{S}_{\text{B14}}$ requires the evaluation of the exponential integral function $\text{Ei}(z)$. This special function is not available in the fortran standard while the erfc function in the primitive of $\mathcal{S}_{\text{B16}}$ has an intrinsic procedure to compute it. This is the reason why we say that "The $\mathcal{S}_{\text{B16}}$ is more adapted" for a Finite-Volume interpretation of the Stokes drift velocity.

- p6,l140–141 This is a nice point. But note on l141 that "summed to $\omega$" should summed with $\omega$. More importantly, please briefly explain how $\omega + \omega^s$ is set; Eq. (10) looks more like a prognostic equation for e3 than a diagnostic equation for $\omega + \omega^s$.
  We changed the wording. For your second remark, the overwhelming majority of NEMO simulations are done with a quasi-Eulerian vertical coordinate (either $z^\star$ or $\sigma$ or a mixture of both) meaning that $\partial_t e_3$ is given by the free-surface evolution (i.e. the coordinate system breathes with the free-surface) and in this context Eq. (10) is used to diagnose $\omega + \omega^s$. With a $\widetilde{z}$-coordinate $\partial_t e_3$ is also prescribed by the evolution of the free-surface but also by the time-evolution of the coordinate

surfaces but the rationale is the same: Eq. (10) is used to diagnose $\omega + \omega^s$. A
sentence has been added to clarify this point.

- p9,l211 Should be $\|\mathbf{u}_{LC}^s\| \propto \sqrt{\|\boldsymbol{\tau}\|}$
  Yes, it is indeed the case, we agree it should be $\sqrt{\|\boldsymbol{\tau}\|}$

- p11,l272–73 "most of the momentum flux going into the waves is quickly trans-
  ferred to the water column through wave breaking "we call this fraction $\tau^{\mathrm{oce}}$". Do
  you mean $\tau_{\mathrm{WW3}}^{\mathrm{oce}}$ ?
  At this point, the notation $\tau^{\mathrm{oce}}$ refers to the momentum flux used as a boundary
  condition for the oceanic model independently from the way it is computed. In
  practice it is indeed in the wave model that $\tau^{\mathrm{oce}}$ is explicitly computed while in
  the oceanic model it is only diagnosed via equation (22). We hope this aspect is
  more clear in the revised manuscript.

- p11,l276 The Charnock number is used to give the surface roughness that pre-
  sumably in fact allows wind to drive waves. Why is the WW3 model then not
  forced by a drag coefficient that includes this effect? Also, why is the stress that
  drives the WW3 waves not stability-dependent?
  This issue has been tackled earlier. The WW3 model is forced by a neutral drag
  coefficient which depends on the Charnock parameter. Historically, the stress
  computation in WW3 does not depend on atmospheric stability because only
  winds were provided to the wave model. As mentioned earlier, we tried to include
  the NEMO bulk formulation in WW3 but the wave solution thus obtained was ex-
  tremely different from the original solution in the neutral case. Changing the bulk
  formulation requires a complete re-calibration of the wave model parameters.
  Again, to our knowledge our practice is customary to all coupled ocean-wave
  models.

- p11,Eq. (21) This equation does seem to ensure that momentum is conserved,
  although I guess $\tau^{\mathrm{atm}} - \tau_{\mathrm{WW3}}^{\mathrm{atm}}$ may be much bigger than it should be.

*Indeed momentum is not conserved because we compute twice the atmospheric flux with two different bulk formulations. As mentioned above, to our knowledge no coupled atmosphere-wave-ocean coupled model guarantees the momentum consistency (on top of the fact that most coupled models use a non-conservative grid-to-grid remapping of the wind-stress). This issue was already explicitly mentioned in the paper in Sec. 3.2: "This strategy is not fully satisfactory since it breaks the momentum conservation".*

- p11,l289–291 I understood that the situation is not that clear, especially for eddy resolving models, and that some consideration does need to be paid to the ocean current when calculating wind stress. E.g. the last sentence of Renault et al. (2018) states: " A simulation without current feedback—by overestimating the eddy amplitude, lifetime, and spatial range."

  *We have to make a distinction depending on the type of coupling with the atmosphere. In a fully coupled mode, oceanic currents have to be taken into account because the corresponding loss of kinetic energy by the ocean is partially compensated by a re-energization of the ocean by the atmospheric PBL. In a forced mode, this re-energization is absent because atmospheric PBL processes are not accounted for and the loss of kinetic energy is thus largely overestimated. Since it is clear that in a forced mode we don't represent the key feedback loops to properly represent the coupling between oceanic currents and the atmosphere we decided not to include this effect. But it should be clear that it is not a limitation of our implementation because it would be straightforward in coupled ocean-wave simulation to include the ocean current when computing the wind-stress (the namelist parameter rn_vfac just needs to be set to $1$ instead of $0$). The objective of our simulations is not to improve our physical understanding of ocean-wave processes but to check the robustness of our implementation.*

- p13 Much of the first para seems to describe the wave model rather than its specific setup, so might fit in better into section 3.1.

The aspects of the wave model discussed in Sec. 4.1.1 are specific to our particular global configuration and other options are available in WW3. That's the reason why we structured it that way. From our point of view Sec. 3.1 should introduce things that are common to any WW3 simulation and necessary to understand where the coupling operates.

- p13"l337–8 "The numerical options are the one commonly chosen by the Drakkar group". This is a bit confusing; please indicate which of the options described here are the Drakkar options, and whether there are other option choices described in the Drakkar website that are not described here.
  In the manuscript we provide most of the information about the options used for the NEMO runs. For more details on those options, the namelist we used for the simulations are available under zenodo. In particular, see
  https://zenodo.org/record/3331463/files/namelist_cfg?download=1 and
  https://zenodo.org/record/3331463/files/namelist_ref?download=1
  Furthermore as the reference to the Drakkar group was indeed confusing since Drakkar refers to NEMO global modelling community and not to specific numerical scheme, this sentence has been removed from the new manuscript

- p13"l342 How does the lateral diffusivity vary away from the equator?
  We have clarified it in the manuscript. The values of lateral (hyper)-viscosity and diffusivity we give in the paper are the values at the equator. Away from the equator those values vary proportionally to $\Delta x$ for the diffusivity and $\Delta x^3$ for the hyper-viscosity.

- p14,l348–351 More specific details and/or references are required here. Is it only solar forcing that is given a diurnal cycle? A reference is required for the data correction to ensure consistency.
  This remark was indeed confusing and was simply wrong. The only correction we make to the forcing fields is to guarantee that their annual mean matches the

annual mean obtained from satellites.

- p14,section4.1.3 There should be a test experiment with ST_CPL + changes to the TKE scheme but no Langmuir parameterization, to see whether the new TKE boundary condition makes any difference.
  This is a good point, we have done this additional experiment (referred to as TKE_CPL) and results are shown in Fig. 10. Note that besides the new boundary condition for TKE other changes have been done also to the boundary condition to diagnose the mixing length and an extra forcing term related to the Stokes drift shear has been added in the TKE equation.

- p14,section4.1.3 Please specify the initial conditions. Is it a spun up run of some standard NEMO setup? If so, give details.
  All ORCA025 experiments have been initialised from reanalysis GLORYS2V4 delivered by MERCATOR-OCEAN-INTERNATIONAL

- p15,Figure2 Various random missing letters on panel titles.
  Yes, the rendering of this figure was fine with Mac but is bad on other operating system. The problem has been solved.

- p16–p17,section4.2.2 Given the amount of space devoted to the extra TKE injection (& 2 figures!), it really does seem strange that no run with $ST_CPL$ + changes to the TKE scheme but no Langmuir parameterization has been presented.
  A new simulation $TKE_CPL$ including all terms of the wave coupling except Langmuir parameterization has been performed and results where added in Figure 10, description on table 2 and discussion added in the text

- p17–p19,section4.2.3 Give reference for ARGO MLD climatology, and specify the MLD criteria used in model and climatology.
  ARGO data are issued from an updated version of de Boyer Montégut et al.
[Figure]

(2004) where the criterion used is Rho_10m-Rho_10m*0.03 . This has been added in the new manuscript

- p17–p19,section4.2.3 Maps of discrepancies of MLD from ARGO, and zonal-average MLDs would be more convincing than the MLD pdfs.
  At first, we looked at maps of discrepancies, but due to the scarcity of the measurements the relevance of such comparison seems meaningless. From our point of view, the best way to compare is to co-localize the model results with the data. Eddies and fronts in the numerical simulations are not at the same place such that it makes more sense to look at PDFs rather than point-by-point differences. That is the reason why it has been chosen to use MLD pdfs. Although far from being an ideal diagnostic it at least shows a reliable statistical improvement of the MLDs when wave coupling is activated.

- p20,445 "an increased heat content during winter leading to higher SST during summer." Is this the wrong way round?
  Indeed this the wrong way round, winter and summer have inverted in the new manuscript

- p29, appendixB It is not easy to see which of these solutions is best. On p9, l25–16, you write " Based on single-column experiments detailed in App. B, we find that parameter values in the range 0.15 - 0.3 provide satisfactory results compared to LES simulations" Where are these LES simulation results?
  The LES results are the one presented in Noh et al. (2016) and our Figure B.1 should be compared to their Fig. 3. We modified the text to clarify this.

---

## Author Comment (AC2) · 17 Feb 2020

**Review from Oyvind Breivik**

First, the authors would like to sincerely thank the reviewers for their careful reading of the paper and their valuable comments to the manuscript and helpful suggestions. We further clarified several issues raised during the review process. Please find attached our revised paper and below a summary of how we responded to the comments. Our comments are reported in color in the text below.

[Figure]

General comments

- This paper describes a two-year experiment with a coupled WW3-NEMO setup. The experiment builds on earlier experiments by Breivik et al (2015) and others who investigated the impact of waves on the mixed layer. The paper is well written and clear.

- First, the change from a Dirichlet to a Neumann condition for the TKE flux should be discussed in more detail. It is not clear to me that comparing against an uncoupled run with a Dirichlet conditon is clean. A separate experiment should be run where the uncoupled model ingests a flux in the Neumann form, or alternatively a coupled run where the Dirichlet condition is used to communicate the TKE flux from WW3. That's a good point. From our point of view as soon as a coupling with a wave model is performed the surface boundary condition should systematically be in the Neumann (flux) form because the wave model naturally provides its information through a flux. In the uncoupled case, it is less clear what should be done. In Mellor and Blumberg (2004)[1] the authors consider both a Dirichlet condition (such that $e_{\text{sfc}} = (15.8\alpha_{\text{CB}})^{2/3}u_\star^2$ (their eq. (10)) ) and an equivalent Neumann condition $(K_e\partial_z e = 2\alpha_{\text{CB}}u_\star^3$ (their eq. (3))) and they mention in their Sec. 7 that *"numerical solutions using Eqs. (1), (2), and "* a Dirichlet condition *"instead of"* a Neumann condition *"reproduced all of the calculations in Figs. 1, 3, and 4"*. Based on their finding, we preferred to focus our efforts in terms of additional simulations toward the clarification of the role of the Axell parameterization on our numerical results. But this would definitely be worth the effort to redo the Mellor & Blumberg experiment with NEMO to check if solutions are indeed insensitive to the nature of the TKE surface boundary condition in uncoupled cases.
* * *
[1]Mellor, G. and A. Blumberg, 2004: Wave Breaking and Ocean Surface Layer Thermal Response. J. Phys. Oceanogr., 34, 693–698.

It should be emphasized that a Neumann (flux) boundary condition for TKE has been used earlier in various studies of wave-ocean coupling (e.g. Michaud et al., 2012) and is not something specific to our approach.

- The integration period is rather short. I think the authors should investigate whether there is sufficient convergence after just two years.
  We are not necessarily looking for convergence but we considered it was enough to illustrate the fact that our developments were actually producing the expected results. Integrating longer in time could also lead to drifts in the stratification independently from the wave effects and could thus distort our interpretation. We are lucid about the fact that we can not draw any conclusion on the long term impact of waves effect at global scales, a different experimental setup would be needed to do that.

- The Langmuir experiment is very interesting as it promises a way forward from the ETAU hack. I would like to see a quantification of how much changing from parameterized Stokes drift (1.6% of the wind speed) to a Stokes drift taken from WW3 gives you. I suspect the most important thing you've done is to change the factor from 0.15 to 0.30. Further on the Langmuir experiment, you don't seem to improve the Stokes drift discussion is interesting.
  Besides the calibration of the parameter $c_{\mathrm{LC}}$ we have also revised the way the input of Stokes drift contributing to Langmuir turbulence is computed. See in Fig. 1 at the end of this document the annual mean of the surface Stokes drift $\|\mathbf{u}^s(\eta)\|$ vs the surface Stokes drift as parameterized in the uncoupled case (i.e. $0.377\sqrt{\|\boldsymbol{\tau}\|/\rho_0}$). On average those two quantities are significantly different which partially explain the stronger role played by the LC parameterization in coupled simulations vs uncoupled ones.

- I suggest you read the appendix of Li et al (2017) where there is a description of the finite volume form of the profile by Breivik et al (2016).

Thanks for pointing this out to us. We now make reference to App. A of Li et
al (2017) when introducing the finite-volume form of the Stokes drift profile. It
seems that Wu et al. (2019) also considered such approach.

- Also, the recent paper by Wu et al (2019) discusses the combined impact (quite
  small!) of the Coriolis-Stokes force and the Stokes drift on tracer advection.
  We were aware of this paper but we forgot to mention it, it has been added to
  the revised manuscript. It is indeed well know that you should have both Stokes-
  Coriolis and the effect of Stokes drift on tracer/continuity equation all together
  otherwise it does not make any sense. Indeed, because of the geostrophic bal-
  ance, the Stokes-Coriolis force must be counterbalanced by a pressure gradient
  which accounts for the Stokes drift. This combined impact seems indeed rather
  small also in our numerical experiments.

- Finally, a quantitative assessment of the relative impact of the various wave-
  induced processes is needed in order to give the reader an idea of their im-
  portance. This applies to the description in Sec 4.2.3 as mentioned below.
  In the revised manuscript an additional numerical experiment has been done to
  further assess the relative role of the different changes (see Tab. 2). This ad-
  ditional simulation allows to separate the impact of the modifications in the TKE
  scheme from the impact of the Langmuir cell parameterization. It suggests that
  for a $1/4°$ resolution the additional terms in the wave-averaged primitive equations
  have very small impact and that most of the improvements we see are related to
  the change in wind-stress, TKE closure and LC parameterization. We believe
  that Fig. 10 provides some hints on the relative role of each processes.

- Cost: You have run WW3 on half the resolution of NEMO at 20% added cost.
  Have you considered the added benefit of running the models on similar resolu-
  tion? I presume this would cost more than twice the standalone NEMO run, so I
  sympathize with your decision, though.

Considering a linear scaling going from $1/2°$ to $1/4°$ would increase the cost of the wave model by $8$ and thus the added cost would be 160%. Besides the associated cost, our study was motivated by operational purposes in the framework of CMEMS, that's the reason why we had to keep a reasonable added cost for the coupling with the waves.

- All told, I would say that after major revision (rerunning the experiments with Dirichlet or Neumann to make a clean comparison) and assessment of the relative importance of the wave-induced effects, this paper should be acceptable for publication in GMD.

  Our study provides in several ways a good starting to allow a clean separation between various effects. We could imagine refining it by implementing online diagnostics to assess the relative role of the different in the prognostic equation for TKE. This is however beyond the scope of this particular study and we think that a $1/4°$ resolution global oceanic configuration is probably not the adequate simulation to do that.

  Moreover, just like the example you give below for the combined impact of Stokes-Coriolis and the Stokes drift in tracer/mass equations the modifications we make are often not independent from each other and trying to test each modification individually may break some balances. We would have liked to prepare a figure showing the difference between $\Phi_{oc}/\rho_0$ (the TKE flux from the wave model in the coupled case) vs $2\alpha_{CB}u_\star^3$ (the TKE flux in the uncoupled case following Craig & Banner) because we did not have enough time to do so because additional experiments would have been needed ($\Phi_{oc}$ was not stored in our standard outputs).

Detailed comments

- Fig 2 is a mess. Please explain in detail what is shown in the different panels and refer to those panels in the text. The figure headings are illegible. I am also
surprised by the huge difference in average wave height and would like to see a more in-depth discussion of why this is so.

Sorry for Fig. 2, the rendering of this figure is perfectly fine on a mac computer but there is something wrong on other operating systems. We have corrected this issue. On this figure we show the seasonal averaged of significant wave height as computed by WW3 on the left panels and the differences between the Charnock parameter computed by WW3 and the standard constant value. Such large deviations of the Charnock parameters from the constant value $0.018$ have also been observed for example by Pineau-Guillou et al. (2018) [2].

- 4.2.2 It is interesting that you have rewritten the Dirichlet conditon to a Neumann condition for the TKE flux. However, I think you should also investigate how this affects the results as you compare against an uncoupled run with a Dirichlet condition.

  Please see our comments on this aspect earlier in our reply.

- 4.2.3 The impact on MLD and SST does not separate between Langmuir, TKE flux, and stress. This needs to be done.

  As mentioned above, an additional numerical experiment has been done to separate those 3 effects and results are shown in Fig. 10.

- I was meant to say about the Langmuir mixing that you don't seem to improve the MLD much, but this is part of the general comment I was making that you need to separate the impact of the various processes.

  Based on the new version of Fig. 10, the effect of the parameterized Langmuir mixing is not significantly less than the effect of the revised TKE scheme. The

[2]Pineau–Guillou, L., Ardhuin, F., Bouin, M.-N., Redelsperger, J.-L., Chapron, B., Bidlot, J.-R. and Quilfen, Y. (2018), Strong winds in a coupled wave–atmosphere model during a North Atlantic storm event: evaluation against observations. Q.J.R. Meteorol. Soc, 144 317-332

figure-1.pdf

**Fig. 1.** Annual average of surface Stokes drift module $\|\mathbf{u}^s(\eta)\|$ (top), of the portion of the Stokes drift aligned with the wind (middle), and of the surface Stokes drift as parameterized by $0.377\sqrt{\|\boldsymbol{\tau}^{\mathrm{oce}}\|/\rho_0}$ in the uncoupled case (bottom)

Axell parameterization is necessary to make the MLD more consistent with observations.

---

## Author Response (AR2)

**Review from Oyvind Breivik**

First, the authors would like to sincerely thank the reviewers for their careful reading of the paper and their valuable comments to the manuscript and helpful suggestions. We further clarified several issues raised during the review process. Please find attached our revised paper and below a summary of how we responded to the comments. Our comments are reported in color in the text below.

**General comments**

- This paper describes a two-year experiment with a coupled WW3-NEMO setup. The experiment builds on earlier experiments by Breivik et al (2015) and others who investigated the impact of waves on the mixed layer. The paper is well written and clear.

- First, the change from a Dirichlet to a Neumann condition for the TKE flux should be discussed in more detail. It is not clear to me that comparing against an uncoupled run with a Dirichlet conditon is clean. A separate experiment should be run where the uncoupled model ingests a flux in the Neumann form, or alternatively a coupled run where the Dirichlet condition is used to communicate the TKE flux from WW3. That's a good point. From our point of view as soon as a coupling with a wave model is performed the surface boundary condition should systematically be in the Neumann (flux) form because the wave model naturally provides its information through a flux. In the uncoupled case, it is less clear what should be done. In Mellor and Blumberg $(2004)^1$ the authors consider both a Dirichlet condition (such that $e_{\text{sfc}} = (15.8\alpha_{\text{CB}})^{2/3}u_\star^2$ (their eq. (10)) ) and an equivalent Neumann condition ($K_e\partial_z e = 2\alpha_{\text{CB}}u_\star^3$ (their eq. (3))) and they mention in their Sec. 7 that *"numerical solutions using Eqs. (1), (2), and "* a Dirichlet condition *"instead of"* a Neumann condition *"reproduced all of the calculations in Figs. 1, 3, and 4"*. Based on their finding, we preferred to focus our efforts in terms of additional simulations toward the clarification of the role of the Axell parameterization on our
* * *
[1]Mellor, G. and A. Blumberg, 2004: Wave Breaking and Ocean Surface Layer Thermal Response. J. Phys. Oceanogr., 34, 693698.

numerical results. But this would definitely be worth the effort to redo the Mellor & Blumberg experiment with NEMO to check if solutions are indeed insensitive to the nature of the TKE surface boundary condition in uncoupled cases. It should be emphasized that a Neumann (flux) boundary condition for TKE has been used earlier in various studies of wave-ocean coupling (e.g. Michaud et al., 2012) and is not something specific to our approach.

- The integration period is rather short. I think the authors should investigate whether there is sufficient convergence after just two years. We are not necessarily looking for convergence but we considered it was enough to illustrate the fact that our developments were actually producing the expected results. Integrating longer in time could also lead to drifts in the stratification independently from the wave effects and could thus distort our interpretation. We are lucid about the fact that we can not draw any conclusion on the long term impact of waves effect at global scales, a different experimental setup would be needed to do that.

- The Langmuir experiment is very interesting as it promises a way forward from the ETAU hack. I would like to see a quantification of how much changing from parameterized Stokes drift (1.6% of the wind speed) to a Stokes drift taken from WW3 gives you. I suspect the most important thing youve done is to change the factor from 0.15 to 0.30. Further on the Langmuir experiment, you dont seem to improve the Stokes drift discussion is interesting.
Besides the calibration of the parameter $c_{\mathrm{LC}}$ we have also revised the way the input of Stokes drift contributing to Langmuir turbulence is computed. See in Fig. 1 at the end of this document the annual mean of the surface Stokes drift $\|\mathbf{u}^s(\eta)\|$ vs the surface Stokes drift as parameterized in the uncoupled case (i.e. $0.377\sqrt{\|\boldsymbol{\tau}\|/\rho_0}$). On average those two quantities are significantly different which partially explain the stronger role played by the LC parameterization in coupled simulations vs uncoupled ones.

- I suggest you read the appendix of Li et al (2017) where there is a description of the finite volume form of the profile by Breivik et al (2016).
Thanks for pointing this out to us. We now make reference to App.

A of Li et al (2017) when introducing the finite-volume form of the Stokes drift profile. It seems that Wu et al. (2019) also considered such approach.

- Also, the recent paper by Wu et al (2019) discusses the combined impact (quite small!) of the Coriolis-Stokes force and the Stokes drift on tracer advection.
We were aware of this paper but we forgot to mention it, it has been added to the revised manuscript. It is indeed well know that you should have both Stokes-Coriolis and the effect of Stokes drift on tracer/continuity equation all together otherwise it does not make any sense. Indeed, because of the geostrophic balance, the Stokes-Coriolis force must be counterbalanced by a pressure gradient which accounts for the Stokes drift. This combined impact seems indeed rather small also in our numerical experiments.

- Finally, a quantitative assessment of the relative impact of the various wave-induced processes is needed in order to give the reader an idea of their importance. This applies to the description in Sec 4.2.3 as mentioned below.
In the revised manuscript an additional numerical experiment has been done to further assess the relative role of the different changes (see Tab. 2). This additional simulation allows to separate the impact of the modifications in the TKE scheme from the impact of the Langmuir cell parameterization. It suggests that for a $1/4°$ resolution the additional terms in the wave-averaged primitive equations have very small impact and that most of the improvements we see are related to the change in wind-stress, TKE closure and LC parameterization. We believe that Fig. 10 provides some hints on the relative role of each processes.

- Cost: You have run WW3 on half the resolution of NEMO at 20% added cost. Have you considered the added benefit of running the models on similar resolution? I presume this would cost more than twice the standalone NEMO run, so I sympathize with your decision, though.
Considering a linear scaling going from $1/2°$ to $1/4°$ would increase the cost of the wave model by 8 and thus the added cost would be 160%. Besides the associated cost, our study was motivated by operational

purposes in the framework of CMEMS, that's the reason why we had to keep a reasonable added cost for the coupling with the waves.

- All told, I would say that after major revision (rerunning the experiments with Dirichlet or Neumann to make a clean comparison) and assessment of the relative importance of the wave-induced effects, this paper should be acceptable for publication in GMD.

  Our study provides in several ways a good starting to allow a clean separation between various effects. We could imagine refining it by implementing online diagnostics to assess the relative role of the different in the prognostic equation for TKE. This is however beyond the scope of this particular study and we think that a $1/4°$ resolution global oceanic configuration is probably not the adequate simulation to do that.

  Moreover, just like the example you give below for the combined impact of Stokes-Coriolis and the Stokes drift in tracer/mass equations the modifications we make are often not independent from each other and trying to test each modification individually may break some balances. We would have liked to prepare a figure showing the difference between $\Phi_{oc}/\rho_0$ (the TKE flux from the wave model in the coupled case) vs $2\alpha_{CB}u_\star^3$ (the TKE flux in the uncoupled case following Craig & Banner) because we did not have enough time to do so because additional experiments would have been needed ($\Phi_{oc}$ was not stored in our standard outputs).

**Detailed comments**

- Fig 2 is a mess. Please explain in detail what is shown in the different panels and refer to those panels in the text. The figure headings are illegible. I am also surprised by the huge difference in average wave height and would like to see a more in-depth discussion of why this is so.

  Sorry for Fig. 2, the rendering of this figure is perfectly fine on a mac computer but there is something wrong on other operating systems. We have corrected this issue. On this figure we show the seasonal averaged of significant wave height as computed by WW3 on the left panels and the differences between the Charnock parameter computed by WW3 and the standard constant value. Such large deviations of

the Charnock parameters from the constant value 0.018 have also been observed for example by Pineau-Guillou et al. (2018) [2].

- 4.2.2 It is interesting that you have rewritten the Dirichlet conditon to a Neumann condition for the TKE flux. However, I think you should also investigate how this affects the results as you compare against an uncoupled run with a Dirichlet condition.
  Please see our comments on this aspect earlier in our reply.

- 4.2.3 The impact on MLD and SST does not separate between Langmuir, TKE flux, and stress. This needs to be done.
  As mentioned above, an additional numerical experiment has been done to separate those 3 effects and results are shown in Fig. 10.

- I was meant to say about the Langmuir mixing that you dont seem to improve the MLD much, but this is part of the general comment I was making that you need to separate the impact of the various processes.
  Based on the new version of Fig. 10, the effect of the parameterized Langmuir mixing is not significantly less than the effect of the revised TKE scheme. The Axell parameterization is necessary to make the MLD more consistent with observations.
* * *
[2]PineauGuillou, L., Ardhuin, F., Bouin, M.N., Redelsperger, J.L., Chapron, B., Bidlot, J.R. and Quilfen, Y. (2018), Strong winds in a coupled waveatmosphere model during a North Atlantic storm event: evaluation against observations. Q.J.R. Meteorol. Soc, 144 317-332

[Figure]

Figure 1: Annual average of surface Stokes drift module $\|\mathbf{u}^s(\eta)\|$ (top), of the portion of the Stokes drift aligned with the wind (middle), and of the surface Stokes drift as parameterized by $0.377\sqrt{\|\boldsymbol{\tau}^{\mathrm{oce}}\|/\rho_0}$ in the uncoupled case (bottom)

**Review from George Nurser**

First, the authors would like to sincerely thank the reviewers for their careful reading of the paper and their valuable comments to the manuscript and helpful suggestions. We further clarified several issues raised during the review process. Please find attached our revised paper and below a summary of how we responded to the comments. Our comments are reported in color in the text below.

**Major Comments**

- The authors have done a lot of work here in producing a coupled version of NEMO with WW3. The explanation of the extra terms added to NEMO is full, very much in the spirit of a GMD contribution, and it is good to see that the code is indeed publicly available.
  Thanks for this encouraging comment, the code is indeed publicly available and the developments are now in the process of being incorporated in the official NEMO release within the H2020 IMMERSE project.

- However, shortcuts have been taken e.g. the use of a neutral drag coefficient independent of Charnock number to estimate the atmospheric stress transferred into the waves, while the total atmospheric stress is separately calculated and depends on Charnock number and atmospheric stability.
  From your remark, it seems that our description of how the surface wind-stress is computed was not clear enough. The computation of the wind-stress in the wave model and in the oceanic model are both function of the Charnock parameter computed by the wave model. As mentionned in Sec. 3.2, in the wave model the general formula used is
  $\boldsymbol{\tau}_{\mathrm{ww3}} = \rho_a C_{\mathrm{DN}} \|\mathbf{u}_{10}^{\mathrm{atm}}\| \mathbf{u}_{10}^{\mathrm{atm}}$ where $C_{\mathrm{DN}} = \left( \frac{\kappa}{\ln\left( \frac{z}{z_0} \right)} \right)^2$ where $z_0$ depends
  on the Charnock parameter. The only difference between $\boldsymbol{\tau}_{\mathrm{ww3}}$ and the stress computed in the oceanic model is that the latter accounts for atmospheric stability. Note that what you call a "shortcut" is what is actually done in all coupled ocean-wave models as none of them guarantees energetic consistency (this point is often swept under the carpet in publications). Let us mention that :

  – Wave models in "forced mode" do not have any information on

> atmospheric temperature/humidity or SST which explains why
> they neglect atmospheric stability in the wind-stress computation.
>
> – The solution of wave models is very sensitive to wind-stress and
>   our wave configuration has been designed and validated in forced
>   mode with neutral drag coefficient. We tried to run WW3 with
>   the same bulk formulation as NEMO but the quality of the wave
>   solution was drastically deteriorated doing so.
>
> We tried to further clarified those aspects in the revised manuscript.

- The testing of the modifications with 2-year runs is rather cursory, but
  I guess that the intention of that short testing period is more to check
  that the code is basically OK rather than to optimize the parameter-
  izations. However, there really should be a test run that includes the
  changes to the TKE model coming from the flux condition using wave
  dissipation energy [(4) above] but not including the Langmuir cell pa-
  rameterization.
  Following your suggestion we have added a new case TKE_CPL in Tab.
  2 which includes all ingredients but the Langmuir Cells parameteriza-
  tion. The results thus obtained are showed and discussed in Fig. 10
  and Sec. 4.2.3.

- The paper generally seems a bit rushed, and the English while being
  perfectly readable, is not great; there are many extra ss where there
  should be none, etc.
  Sorry for that, we tried to correct as much as possible these issues.

**Detailed Comments**

- p2,l39 Should refer to Lu et al. (2019).
  We believe that instead of Lu et al. (2019) it should be Wu et al. (2019)
  ? A reference to Wu et al. (2019) has been added, in particular they
  also compute the Stokes drift as a "layer-averaged Stokes drift profile"
  based on the Breivik et al. (2016) profile. It is really not clear from
  their paper but it seems that they introduced in NEMO a Neumann
  (flux) boundary condition for the TKE equation. Thanks for pointing
  out this reference to us.

- p3,Eqs. (1)(4) Various w should be $\omega$.
  It has been corrected in subgrid scale terms

- p3,Eqs.(1)(4) Please define $p_h$ and $p_s$.
  Done

- p3,l78 $\tau^{\text{oce}}$ is not strictly the wind stress; it is that part of the stress that drives the ocean rather than developing the wave field.
  Indeed you are right, it has been changed.

- p3,l78 Please explain the dynamic boundary condition imposing the continuity of pressure at the air-sea interface
  We do not necessarily see what should be explained here. There must be no pressure jump at the air-sea interface, continuity of pressure translates into $p = p_{\text{atm}}$ at the interface with $p_{\text{atm}}$ the atmospheric pressure at sea surface.

- p3,l80 $\omega(z = -H) = 0$. This is only true for terrain following coordinates, not for a generalized coordinate.
  Since $\omega$ is the dia-surface velocity component, at the lower boundary the no-normal flow boundary condition should read $\omega_{\text{bot}} = 0$ which is equivalent to $\mathbf{u} \cdot \mathbf{n} = 0$. We do not understand the issue here, the continuity equation equation integrates starting from $\omega = 0$ even with geopotential coordinate.

- p4,Eqs.(7)(8) Please define $p^{\text{J}}$ and $p^{\text{FV}}$. I assume $p^{\text{J}}$ is the J that is only significant in shallow water, defined in eq. (20) of Bennis et al., (2011). If so, then presumably $p^{\text{FV}}$ represents the term $\frac{1}{2}\left[(\mathbf{u} + \mathbf{u}^s)^2 - \mathbf{u}^2\right]$ found e.g. on the RHS of Eq. (2) of Suzuki and Fox-Kemper (2016). This term would seem to scale with the vortex force term. Can the authors justify its neglect?
  Thanks for raising this issue. First we tried to clarify the notations in the paper and the way the additional wave related terms are introduced. In Suzuki and Fox-Kemper (2016) (SFK16) the Craik-Leibovich (CL) equations are used while our implementation relies on the more general wave-averaged primitive equations. In the CL equations the Bernoulli head term (let us note it $\mathcal{K}$) is defined as the kinetic energy increase due to the waves, i.e. $\mathcal{K} = \dfrac{\|\mathbf{u}^s + \mathbf{u}\|^2}{2} - \dfrac{\|\mathbf{u}\|^2}{2}$. In the wave-averaged equations, the form of the Bernoulli head is much more complicated

(see eq (9.20) in McWilliams et al. 2004 or the $S^{\text{Shear}}$ term in Eq. (40) in Ardhuin et al. 2008) and it does not appear explicitly in our implementation because of the general weak vertical shears in the wave-mixed layer. The effect of that term was also found to be much weaker than $S^J$ in shallow coastal environments, except in the surf zone. It is also mentioned in SFK16 (their Eq (14)) that the contribution of the Stokes shear force should be retained but since this term results from the combination of the vertical component of the vortex force with the Bernoulli head, it does not appear explicitly in our derivation. Finally, compared to the previous version of the manuscript, additional terms related to the slope of the vertical coordinate have been added in Eqs. (7) and (8). Those terms are pieces of the vortex force which need to be taken into account with a generalized vertical coordinate. They seldom appear in the literature because most people present the wave-averaged equations in geopotential coordinate.

- p4,Eq.(12) The wave pressure vector seems a little odd. It would be more natural to make $\mathcal{W}_{\text{prs}}$ the column vector of x- y- and vertical gradients, especially given that in p5, l118 you refer to the additional wave-induced barotropic forcing terms corresponding to the vertical integral of the ... $\mathcal{W}_{\text{prs}}$
  This has been reformulated by including the gradients in the $\mathcal{W}_{\text{prs}}$ term and removing the reference to the vertical integral of $\mathcal{W}_{\text{prs}}$ since after our simplifications this becomes a 2D horizontal field. The notations have been adapted and are now more consistent with Bennis et al. (2011) and Michaud et al. (2012) except that the $S^J$ and $S^{\text{Shear}}$ terms are expressed directly in terms of pressure terms $\widetilde{p}^J$ and $\widetilde{p}^{\text{Shear}}$.

- p5,Eq.(13) How do you decompose baroclinic and barotropic contributions to bottom drag when using non-linear bottom drag, as you do in these experiments (p13, l342) ?
  The non-linear bottom drag in the baroclinic mode is computed in an implicit way as $(C_D\|\mathbf{u}_h\|)^{n-1}\mathbf{u}_h^{n+1}$. Because of the linearization this term is analogous to a linear bottom drag and thus easy to separate into a barotropic and a baroclinic contributions. See Sec. 10.4 in the version 4.0 of the NEMO documentation for more details.

- p6,Figure1 You seem to have evaluated the primitive $\mathcal{I}_{\text{B14}}$,as you plot it out here. Why is it not written out explicitly like $\mathcal{I}_{\text{B16}}$, which is set

out at the bottom of p5?

The primitive of $\mathcal{S}_{\mathrm{B}14}$ requires the evaluation of the exponential integral function $\mathrm{Ei}(z)$. This special function is not available in the fortran standard while the erfc function in the primitive of $\mathcal{S}_{\mathrm{B}16}$ has an intrinsic procedure to compute it. This is the reason why we say that "The $\mathcal{S}_{\mathrm{B}16}$ is more adapted" for a Finite-Volume interpretation of the Stokes drift velocity.

- p6,l140141 This is a nice point. But note on l141 that summed to $\omega$ should summed with $\omega$. More importantly, please briefly explain how $\omega + \omega^s$ is set; Eq. (10) looks more like a prognostic equation for e3 than a diagnostic equation for $\omega + \omega^s$.

  We changed the wording. For your second remark, the overwhelming majority of NEMO simulations are done with a quasi-Eulerian vertical coordinate (either $z^\star$ or $\sigma$ or a mixture of both) meaning that $\partial_t \mathrm{e}_3$ is given by the free-surface evolution (i.e. the coordinate system breathes with the free-surface) and in this context Eq. (10) is used to diagnose $\omega + \omega^s$. With a $\widetilde{z}$-coordinate $\partial_t \mathrm{e}_3$ is also prescribed by the evolution of the free-surface but also by the time-evolution of the coordinate surfaces but the rationale is the same: Eq. (10) is used to diagnose $\omega + \omega^s$. A sentence has been added to clarify this point.

- p9,l211 Should be $\|\mathbf{u}_{\mathrm{LC}}^s\| \propto \sqrt{\|\boldsymbol{\tau}\|}$

  Yes, it is indeed the case, we agree it should be $\sqrt{\|\boldsymbol{\tau}\|}$

- p11,l27273 most of the momentum flux going into the waves is quickly transferred to the water column through wave breaking "we call this fraction $\boldsymbol{\tau}^{\mathrm{oce}}$. Do you mean $\boldsymbol{\tau}_{\mathrm{WW3}}^{\mathrm{oce}}$ ?

  At this point, the notation $\boldsymbol{\tau}^{\mathrm{oce}}$ refers to the momentum flux used as a boundary condition for the oceanic model independently from the way it is computed. In practice it is indeed in the wave model that $\boldsymbol{\tau}^{\mathrm{oce}}$ is explicitly computed while in the oceanic model it is only diagnosed via equation (22). We hope this aspect is more clear in the revised manuscript.

- p11,l276 The Charnock number is used to give the surface roughness that presumably in fact allows wind to drive waves. Why is the WW3 model then not forced by a drag coefficient that includes this

effect? Also, why is the stress that drives the WW3 waves not stability-dependent?

This issue has been tackled earlier. The WW3 model is forced by a neutral drag coefficient which depends on the Charnock parameter. Historically, the stress computation in WW3 does not depend on atmospheric stability because only winds were provided to the wave model. As mentioned earlier, we tried to include the NEMO bulk formulation in WW3 but the wave solution thus obtained was extremely different from the original solution in the neutral case. Changing the bulk formulation requires a complete re-calibration of the wave model parameters. Again, to our knowledge our practice is customary to all coupled ocean-wave models.

- p11,Eq. (21) This equation does seem to ensure that momentum is conserved, although I guess $\boldsymbol{\tau}^{\mathrm{atm}} - \boldsymbol{\tau}_{\mathrm{WW3}}^{\mathrm{atm}}$ may be much bigger than it should be.

  Indeed momentum is not conserved because we compute twice the atmospheric flux with two different bulk formulations. As mentioned above, to our knowledge no coupled atmosphere-wave-ocean coupled model guarantees the momentum consistency (on top of the fact that most coupled models use a non-conservative grid-to-grid remapping of the wind-stress). This issue was already explicitly mentioned in the paper in Sec. 3.2: "*This strategy is not fully satisfactory since it breaks the momentum conservation*".

- p11,l289291 I understood that the situation is not that clear, especially for eddy resolving models, and that some consideration does need to be paid to the ocean current when calculating wind stress. E.g. the last sentence of Renault et al. (2018) states: A simulation without current feedbackby overestimating the eddy amplitude, lifetime, and spatial range.

  We have to make a distinction depending on the type of coupling with the atmosphere. In a fully coupled mode, oceanic currents have to be taken into account because the corresponding loss of kinetic energy by the ocean is partially compensated by a re-energization of the ocean by the atmospheric PBL. In a forced mode, this re-energization is absent because atmospheric PBL processes are not accounted for and the loss of kinetic energy is thus largely overestimated. Since it is clear that in

a forced mode we don't represent the key feedback loops to properly represent the coupling between oceanic currents and the atmosphere we decided not to include this effect. But it should be clear that it is not a limitation of our implementation because it would be straightforward in coupled ocean-wave simulation to include the ocean current when computing the wind-stress (the namelist parameter rn_vfac just needs to be set to 1 instead of 0). The objective of our simulations is not to improve our physical understanding of ocean-wave processes but to check the robustness of our implementation.

- p13 Much of the first para seems to describe the wave model rather than its specific setup, so might fit in better into section 3.1.
  The aspects of the wave model discussed in Sec. 4.1.1 are specific to our particular global configuration and other options are available in WW3. That's the reason why we structured it that way. From our point of view Sec. 3.1 should introduce things that are common to any WW3 simulation and necessary to understand where the coupling operates.

- p13,,l3378 The numerical options are the one commonly chosen by the Drakkar group. This is a bit confusing; please indicate which of the options described here are the Drakkar options, and whether there are other option choices described in the Drakkar website that are not described here.
  In the manuscript we provide most of the information about the options used for the NEMO runs. For more details on those options, the namelist we used for the simulations are available under zenodo. In particular, see
  https://zenodo.org/record/3331463/files/namelist˙cfg?download=1 and https://zenodo.org/record/3331463/files/namelist˙ref?download=1
  Furthermore as the reference to the Drakkar group was indeed confusing since Drakkar refers to NEMO global modelling community and not to specific numerical scheme, this sentence has been removed from the new manuscript

- p13,,l342 How does the lateral diffusivity vary away from the equator?
  We have clarified it in the manuscript. The values of lateral (hyper)-viscosity and diffusivity we give in the paper are the values at the

equator. Away from the equator those values vary proportionally to $\Delta x$ for the diffusivity and $\Delta x^3$ for the hyper-viscosity.

- p14,l348351 More specific details and/or references are required here. Is it only solar forcing that is given a diurnal cycle? A reference is required for the data correction to ensure consistency.
  This remark was indeed confusing and was simply wrong. The only correction we make to the forcing fields is to guarantee that their annual mean matches the annual mean obtained from satellites.

- p14,section4.1.3 There should be a test experiment with ST_CPL + changes to the TKE scheme but no Langmuir parameterization, to see whether the new TKE boundary condition makes any difference.
  This is a good point, we have done this additional experiment (referred to as TKE_CPL) and results are shown in Fig. 10. Note that besides the new boundary condition for TKE other changes have been done also to the boundary condition to diagnose the mixing length and an extra forcing term related to the Stokes drift shear has been added in the TKE equation.

- p14,section4.1.3 Please specify the initial conditions. Is it a spun up run of some standard NEMO setup? If so, give details.
  All ORCA025 experiments have been initialised from reanalysis GLO-RYS2V4 delivered by MERCATOR-OCEAN-INTERNATIONAL

- p15,Figure2 Various random missing letters on panel titles.
  Yes, the rendering of this figure was fine with Mac but is bad on other operating system. The problem has been solved.

- p16p17,section4.2.2 Given the amount of space devoted to the extra TKE injection (& 2 figures!), it really does seem strange that no run with ST˙CPL + changes to the TKE scheme but no Langmuir parameterization has been presented.
  A new simulation "TKE˙CPL" including all terms of the wave coupling except Langmuir parameterization has been performed and results where added in Figure 10, description on table 2 and discussion added in the text

- p17p19,section4.2.3 Give reference for ARGO MLD climatology, and specify the MLD criteria used in model and climatology.

ARGO data are issued from an updated version of de Boyer Montgut et al. (2004) where the criterion used is Rho_10m-Rho_10m*0,03. This has been added in the new manuscript

- p17p19,section4.2.3 Maps of discrepancies of MLD from ARGO, and zonal-average MLDs would be more convincing than the MLD pdfs.
  At first, we looked at maps of discrepancies, but due to the scarcity of the measurements the relevance of such comparison seems meaningless. From our point of view, the best way to compare is to co-localize the model results with the data. Eddies and fronts in the numerical simulations are not at the same place such that it makes more sense to look at PDFs rather than point-by-point differences. That is the reason why it has been chosen to use MLD pdfs. Although far from being an ideal diagnostic it at least shows a reliable statistical improvement of the MLDs when wave coupling is activated.

- p20,445 an increased heat content during winter leading to higher SST during summer. Is this the wrong way round?
  Indeed this the wrong way round, winter and summer have inverted in the new manuscript

- p29, appendixB It is not easy to see which of these solutions is best. On p9, l2516, you write  Based on single-column experiments detailed in App. B, we find that parameter values in the range 0.15 - 0.3 provide satisfactory results compared to LES simulations Where are these LES simulation results?
  The LES results are the one presented in Noh et al. (2016) and our Figure B.1 should be compared to their Fig. 3. We modified the text to clarify this.

[revised manuscript text omitted]

$$\partial_k\left(p_h + \widetilde{p}^{\mathrm{Shear}}\right) = -\rho g e_3 \underline{-\partial_k{}^{\mathrm{FV}}} + \rho_0\left(\widetilde{u}^s\partial_k u + \widetilde{v}^s\partial_k v\right)$$

where wave-induced terms are represented with tildes. The $\underline{\widetilde{F}^u \text{ and } \widetilde{F}^v}$ terms represent the sink/source of wave-momentum due to breaking, bottom friction and wave-turbulence interaction. These terms will be neglected since they are expected to play a significant role only in the surf zone. The other extra contributions to the momentum equations include the Stokes-Coriolis force $\mathcal{W}_{\mathrm{St-Cor}}$, the vortex force $\mathcal{W}_{\mathrm{VF}}$, and a wave-induced pressure $\mathcal{W}_{\mathrm{Prs}}$

$$\mathcal{W}_{\mathrm{St-Cor}} = \begin{pmatrix} f\widetilde{v}^s \\ -f\widetilde{u}^s \\ 0 \end{pmatrix}, \qquad \mathcal{W}_{\mathrm{VF}} = \begin{pmatrix} \zeta\widetilde{v}^s - \frac{\widetilde{\omega}^s}{e_3}\partial_k u \\ -\zeta\widetilde{u}^s - \frac{\widetilde{\omega}^s}{e_3}\partial_k v \\ \frac{\widetilde{u}^s}{e_3}\partial_k u + \frac{\widetilde{v}^s}{e_3}\partial_k v \end{pmatrix}, \qquad \mathcal{W}_{\mathrm{Prs}} = \begin{pmatrix} \widetilde{p}^J + \widetilde{p}^{\mathrm{FV}} \\ \widetilde{p}^J + \widetilde{p}^{\mathrm{FV}} \\ \widetilde{p}^{\mathrm{FV}} \end{pmatrix}$$

$$\boldsymbol{\mathcal{W}}_{\text{St}-\text{Cor}} = \begin{pmatrix} f\widetilde{v}^s \\ -f\widetilde{u}^s \\ 0 \end{pmatrix}, \quad \boldsymbol{\mathcal{W}}_{\text{VF}} = \begin{pmatrix} \zeta\widetilde{v}^s - \frac{\widetilde{\omega}^s}{e_3}\partial_k u - \frac{(\partial_x z)}{e_3}\left(\widetilde{u}^s\partial_k u + \widetilde{v}^s\partial_k v\right) \\ -\zeta\widetilde{u}^s - \frac{\widetilde{\omega}^s}{e_3}\partial_k v - \frac{(\partial_y z)}{e_3}\left(\widetilde{u}^s\partial_k u + \widetilde{v}^s\partial_k v\right) \\ \frac{\widetilde{u}^s}{e_3}\partial_k u + \frac{\widetilde{v}^s}{e_3}\partial_k v \end{pmatrix},$$

$$\boldsymbol{\mathcal{W}}_{\text{Prs}} = -\frac{1}{\rho_0}\begin{pmatrix} \partial_x\left(\widetilde{p}^J + \widetilde{p}^{\text{Shear}}\right) - (\partial_x z)\frac{\partial_k(\widetilde{p}^{\text{Shear}})}{e_3} \\ \partial_y\left(\widetilde{p}^J + \widetilde{p}^{\text{Shear}}\right) - (\partial_y z)\frac{\partial_k(\widetilde{p}^{\text{Shear}})}{e_3} \\ \frac{1}{e_3}\partial_k\widetilde{p}^{\text{Shear}} \end{pmatrix} \qquad (12)$$

where the terms involving horizontal derivatives of $\omega$ have been neglected in $\boldsymbol{\mathcal{W}}_{\text{VF}}$. In $\boldsymbol{\mathcal{W}}_{\text{Prs}}$, the $\widetilde{p}^J$ term corresponds to a depth uniform wave-induced kinematic pressure term, while $\widetilde{p}^{\text{FV}}$[1], while $\widetilde{p}^{\text{Shear}}$ is a shear-induced three-dimensional pressure term[2] associated with the vertical component of the vortex force. The vortex force contribution $\boldsymbol{\mathcal{W}}_{\text{VF}}$ can be further simplified by neglecting the terms involving the vertical shear as in Bennis et al. (2011), thus leading to $\boldsymbol{\mathcal{W}}_{\text{VF}} \cdot (0,0,1)^t = 0$ and $\widetilde{p}^{\text{FV}} = 0$.

In particular, the vertical component of the vortex force is absorbed in a pressure term $\widetilde{p}^{\text{Shear}}$ (that gives the $S^{\text{shear}}$ term in the notations of A . 
[revised manuscript text omitted]
_{\mathrm{LC}}$ their expected depth. Following Axell (2002), $w_{\mathrm{LC}}$ and $d_{\mathrm{LC}}$ are given by

$$w_{\mathrm{LC}} = \begin{cases} c_{\mathrm{LC}}\|\widehat{\mathbf{u}}_{\mathrm{LC}}^s\|\sin\left(-\frac{\pi z}{d_{\mathrm{LC}}}\right), & \text{if } -z \le d_{\mathrm{LC}} \\ 0, & \text{otherwise} \end{cases} , \qquad -\int_{-d_{\mathrm{LC}}}^{\eta} N^2(z)z\,dz = \frac{\|\widehat{\mathbf{u}}_{\mathrm{LC}}^s\|^2}{2}$$

where $\|\widehat{\mathbf{u}}_{\mathrm{LC}}^s\|$ is the portion of the surface Stokes drift contributing to Langmuir cells intensity and $c_{\mathrm{LC}}$ a constant parameter. In the absence of information about the wave field it is generally assumed that $\|\widehat{\mathbf{u}}_{\mathrm{LC}}^s\| \propto \|\boldsymbol{\tau}\|$ $\|\widehat{\mathbf{u}}_{\mathrm{LC}}^s\| \propto \sqrt{\|\boldsymbol{\tau}^{\mathrm{oce}}\|}$. 
[revised manuscript text omitted]

---

## Author Response (AR3)

**Topical Editor comments**

Dear authors,
Some minor revisions are required before the paper will be accepted. See the comments from the reviewer.
Best wishes
Qiang Wang

The authors would like to thank the editor Qiang Wang for his valuable help during the review process. Again, we apologize for the inconvenience caused by our careless mistake when submitting the first revision of the manuscript. Please find attached our revised manuscript and a summary of the changes.

There is one equation which I believe the authors must check before the paper is acceptable for publication. On Line 180 is says that $u_s\partial_z = \partial_z u_s + \partial_z(u_s)$. I believe the sign in front of the first term on the RHS should be negative. How has this affected your implementation?

There was indeed a sign error in the equation on line 180, thanks for noticing it. Fortunately, this error does not interfere with our implementation because what is truly implemented in the code is equation (14) which is correct.

**Minor corrections**

- l 45: Not Van Roekel, but Valcke 2012, I believe

- l 115: weater → weaker

- l 195: $S_{\mathrm{oce}}$? Should it not read $S_{\mathrm{ds}}$?

- Fig 2: The lowest panel has a title US=f(tau) which sits atop the numbers 600 and 800. A little more space is neeed to make it more readable. I don't think this is a problem with the rendering as the authors claim - I also use a Mac.

- l 295: rugosity → roughness?

- l 434: reminder → remainder

- Fig 9: southern and northern hemisphere, not south and north

- l 464: retroactions → interaction?

Thanks for the careful reading, all remarks have been taken into account. Regarding Fig. 2 and Fig. 3 we have redesigned those figures to improve their interpretation.

[revised manuscript text omitted]